# A Generalist Agent

**Scott Reed**[*,†], **Konrad Żołna**[*], **Emilio Parisotto**[*], **Sergio Gómez Colmenarejo**[†], **Alexander Novikov**,
**Gabriel Barth-Maron, Mai Giménez, Yury Sulsky, Jackie Kay, Jost Tobias Springenberg, Tom Eccles,
Jake Bruce, Ali Razavi, Ashley Edwards, Nicolas Heess, Yutian Chen, Raia Hadsell, Oriol Vinyals,
Mahyar Bordbar** and **Nando de Freitas**[†]

[*]Equal contributions, [†]Equal senior contributions, All authors are affiliated with DeepMind    *reedscot@deepmind.com*

Reviewed on OpenReview: https://openreview.net/forum?id=1ikK0kHjvj

## Abstract

Inspired by progress in large-scale language modeling, we apply a similar approach towards building a single generalist agent beyond the realm of text outputs. The agent, which we refer to as Gato, works as a multi-modal, multi-task, multi-embodiment generalist policy. The same network with the same weights can play Atari, caption images, chat, stack blocks with a real robot arm and much more, deciding based on its context whether to output text, joint torques, button presses, or other tokens. In this report we describe the model and the data, and document the current capabilities of Gato.

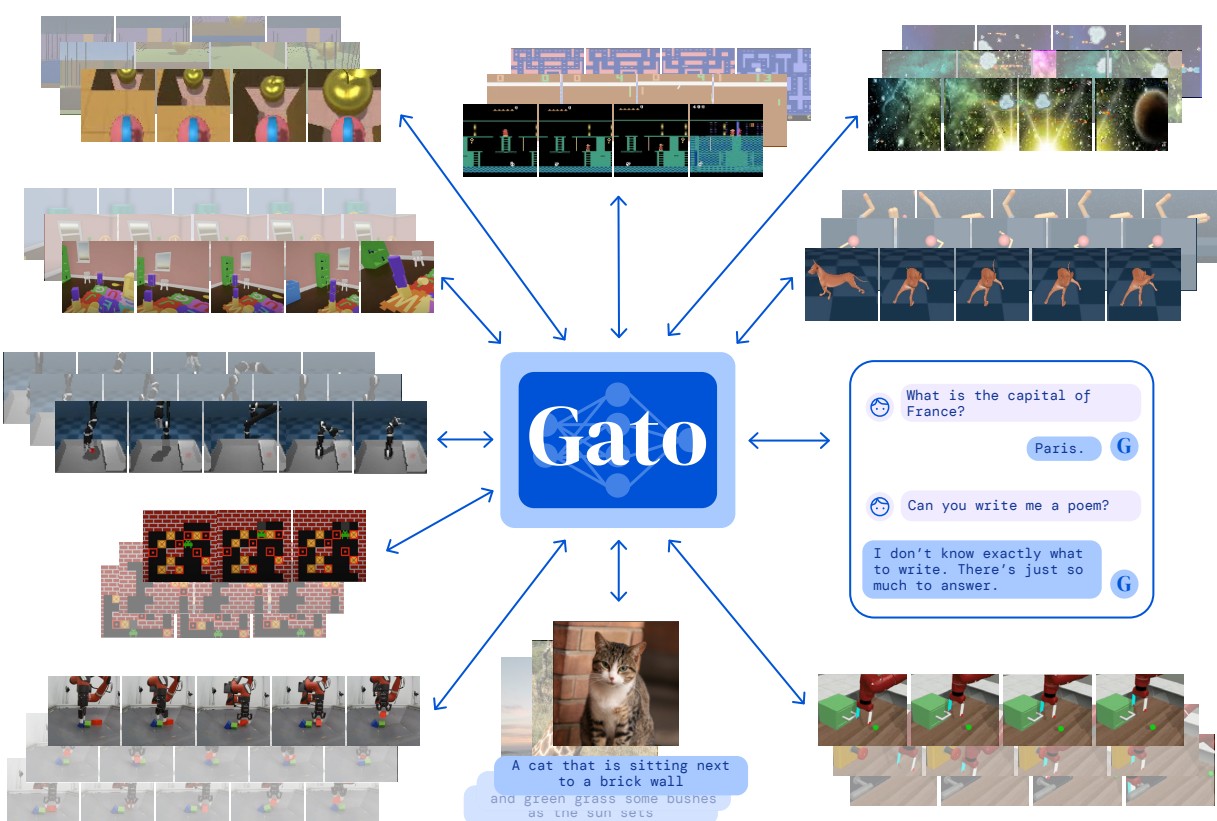

Figure 1: **A generalist agent.** Gato can sense and act with different embodiments across a wide range of environments using a single neural network with the same set of weights. Gato was trained on 604 distinct tasks with varying modalities, observations and action specifications.

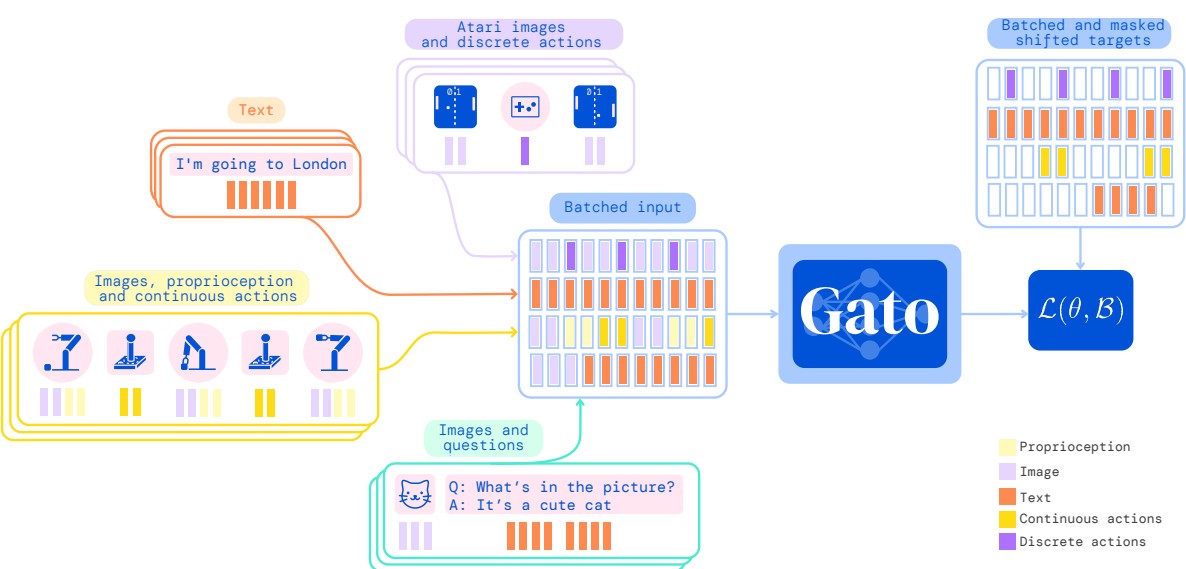

Figure 2: **Training phase of Gato**. Data from different tasks and modalities is serialized into a flat sequence of tokens, batched, and processed by a transformer neural network akin to a large language model. Masking is used such that the loss function is applied only to target outputs, i.e. text and various actions.

# 1 Introduction

There are significant benefits to using a single neural sequence model across all tasks. It reduces the need for hand crafting policy models with appropriate inductive biases for each domain. It increases the amount and diversity of training data since the sequence model can ingest any data that can be serialized into a flat sequence. Furthermore, its performance continues to improve even at the frontier of data, compute and model scale (Kaplan et al., 2020; Hoffmann et al., 2022). Historically, generic models that are better at leveraging computation have also tended to overtake more specialized domain-specific approaches (Sutton, 2019), eventually.

In this paper, we describe the current iteration of a general-purpose agent which we call Gato, instantiated as a single, large, transformer sequence model. With a single set of weights, Gato can engage in dialogue, caption images, stack blocks with a real robot arm, outperform humans at playing Atari games, navigate in simulated 3D environments, follow instructions, and more.

While no agent can be expected to excel in all imaginable control tasks, especially those far outside of its training distribution, we here test the hypothesis that training an agent which is generally capable on a *large number* of tasks is possible; and that this general agent can be adapted with little extra data to succeed at an even larger number of tasks. We hypothesize that such an agent can be obtained through scaling data, compute and model parameters, continually broadening the training distribution while maintaining performance, towards covering any task, behavior and embodiment of interest. In this setting, natural language can act as a common grounding across otherwise incompatible embodiments, unlocking combinatorial generalization to new behaviors.

We focus our training at the operating point of model scale that allows real-time control of real-world robots, currently around 1.2B parameters in the case of Gato. As hardware and model architectures improve, this operating point will naturally increase the feasible model size, pushing generalist models higher up the scaling law curve. For simplicity Gato was trained offline in a purely supervised manner; however, in principle, there is no reason it could not also be trained with either offline or online reinforcement learning (RL).

## 2 Model

The guiding design principle of Gato is to train on the widest variety of relevant data possible, including diverse modalities such as images, text, proprioception, joint torques, button presses, and other discrete and continuous observations and actions. To enable processing this multi-modal data, we serialize all data into a flat sequence of tokens. In this representation, Gato can be trained and sampled from akin to a standard large-scale language model. During deployment, sampled tokens are assembled into dialogue responses, captions, button presses, or other actions based on the context. In the following subsections, we describe Gato's tokenization, network architecture, loss function, and deployment.

### 2.1 Tokenization

There are infinite possible ways to transform data into tokens, including directly using the raw underlying byte stream. Below we report the tokenization scheme we found to produce the best results for Gato at the current scale using contemporary hardware and model architectures.

- Text is encoded via SentencePiece (Kudo & Richardson, 2018) with 32000 subwords into the integer range $[0, 32000)$.

- Images are first transformed into sequences of non-overlapping $16 \times 16$ patches in raster order, as done in ViT (Dosovitskiy et al., 2020). Each pixel in the image patches is then normalized between $[-1, 1]$ and divided by the square-root of the patch size (i.e. $\sqrt{16} = 4$).

- Discrete values, e.g. Atari button presses, are flattened into sequences of integers in row-major order. The tokenized result is a sequence of integers within the range of $[0, 1024)$.

- Continuous values, e.g. proprioceptive inputs or joint torques, are first flattened into sequences of floating point values in row-major order. The values are mu-law encoded to the range $[-1, 1]$ if not already there (see Figure 14 for details), then discretized to 1024 uniform bins. The discrete integers are then shifted to the range of $[32000, 33024)$.

After converting data into tokens, we use the following canonical sequence ordering.

- Text tokens in the same order as the raw input text.

- Image patch tokens in raster order.

- Tensors in row-major order.

- Nested structures in lexicographical order by key.

- Agent timesteps as observation tokens followed by a separator, then action tokens.

- Agent episodes as timesteps in time order.

Further details on tokenizing agent data are presented in the supplementary material (Section B).

### 2.2 Embedding input tokens and setting output targets

After tokenization and sequencing, we apply a parameterized embedding function $f(\cdot; \theta_e)$ to each token (i.e. it is applied to both observations and actions) to produce the final model input. To enable efficient learning from our multi-modal input sequence $s_{1:L}$ the embedding function performs different operations depending on the modality the token stems from:

- Tokens belonging to text, discrete- or continuous-valued observations or actions for any time-step are embedded via a lookup table into a learned vector embedding space. Learnable position encodings are added for all tokens based on their local token position within their corresponding time-step.

- Tokens belonging to image patches for any time-step are embedded using a single ResNet (He et al., 2016a) block to obtain a vector per patch. For image patch token embeddings, we also add a learnable within-image position encoding vector.

We refer to appendix Section C.3 for full details on the embedding function.

As we model the data autoregressively, each token is potentially also a target label given the previous tokens. Text tokens, discrete and continuous values, and actions can be directly set as targets after tokenization. Image tokens and agent nontextual observations are not currently predicted in Gato, although that may be an interesting direction for future work. Targets for these non-predicted tokens are set to an unused value and their contribution to the loss is masked out.

## 2.3 Training

Given a sequence of tokens $s_{1:L}$ and parameters $\theta$, we model the data using the chain rule of probability:

$$\log p_\theta(s_1, \ldots, s_L) = \sum_{l=1}^{L} \log p_\theta(s_l | s_1, \ldots, s_{l-1}), \tag{1}$$

Let $b$ index a training batch of sequences $\mathcal{B}$. We define a masking function $m$ such that $m(b, l) = 1$ if the token at index $l$ is either from text or from the logged action of an agent, and 0 otherwise. The training loss for a batch $\mathcal{B}$ can then be written as

$$\mathcal{L}(\theta, \mathcal{B}) = -\sum_{b=1}^{|\mathcal{B}|} \sum_{l=1}^{L} m(b, l) \log p_\theta \left( s_l^{(b)} | s_1^{(b)}, \ldots, s_{l-1}^{(b)} \right) \tag{2}$$

As described above, Gato's network architecture has two main components: the parameterized embedding function which transforms tokens to token embeddings, and the sequence model which outputs a distribution over the next discrete token. While any general sequence model can work for next token prediction, we chose a transformer (Vaswani et al., 2017) for simplicity and scalability. Gato uses a 1.2B parameter decoder-only transformer with 24 layers, an embedding size of 2048, and a post-attention feedforward hidden size of 8196 (more details in Section C.1).

Because distinct tasks within a domain can share identical embodiments, observation formats and action specifications, the model sometimes needs further context to disambiguate tasks. Rather than providing e.g. one-hot task identifiers, we instead take inspiration from (Sanh et al., 2022; Wei et al., 2021; Brown et al., 2020) and use prompt conditioning. During training, for 25% of the sequences in each batch, a prompt sequence is prepended, coming from an episode generated by the same source agent on the same task. Half of the prompt sequences are from the end of the episode, acting as a form of goal conditioning for many domains; and the other half are uniformly sampled from the episode. During evaluation, the agent can be prompted using a successful demonstration of the desired task, which we do by default in all control results that we present here.

Training of the model is performed on a 16x16 TPU v3 slice for 1M steps with batch size 512 and token sequence length $L = 1024$, which takes about 4 days. Architecture details can be found in Section C. Because agent episodes and documents can easily contain many more tokens than fit into context, we randomly sample subsequences of $L$ tokens from the available episodes. Each batch mixes subsequences approximately uniformly over domains (e.g. Atari, MassiveWeb, etc.), with some manual upweighting of larger and higher quality datasets (see Table 1 in Section 3 for details).

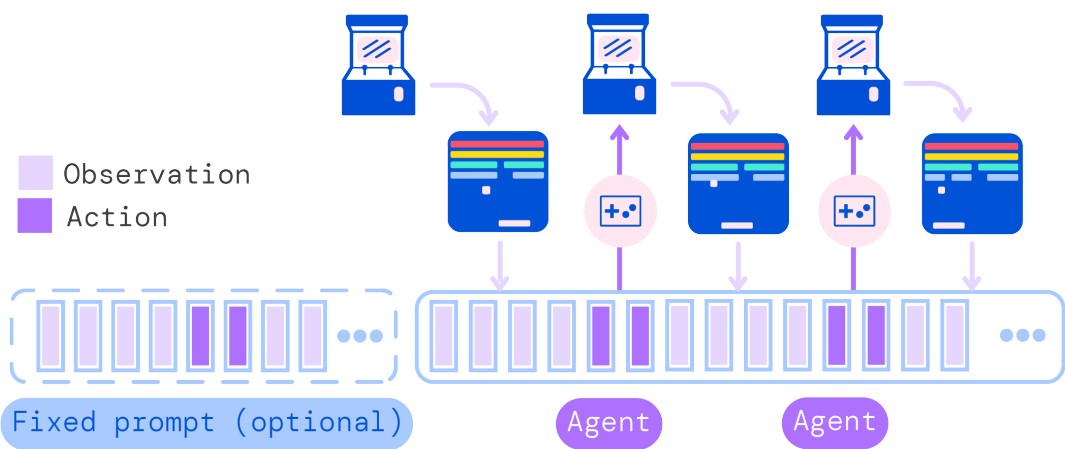

Figure 3: **Running Gato as a control policy.** Gato consumes a sequence of interleaved tokenized observations, separator tokens, and previously sampled actions to produce the next action in standard autoregressive manner. The new action is applied to the environment – a game console in this illustration, a new set of observations is obtained, and the process repeats.

## 2.4 Deployment

Deploying Gato as a policy is illustrated in Figure 3. First a prompt, such as a demonstration, is tokenized, forming the initial sequence. By default, we take the first 1024 tokens of the demonstration. Next the environment yields the first observation which is tokenized and appended to the sequence. Gato samples the action vector autoregressively one token at a time. Once all tokens comprising the action vector have been sampled (determined by the action specification of the environment), the action is decoded by inverting the tokenization procedure described in Section 2.1. This action is sent to the environment which steps and yields a new observation. The procedure repeats. The model always sees all previous observations and actions in its context window of 1024 tokens. We found it beneficial to use transformer XL memory during deployment, although it was not used during training (Dai et al., 2019).

## 3 Datasets

Gato is trained on a large number of datasets comprising agent experience in both simulated and real world environments, as well as a variety of natural language and image datasets. The datasets we use and their attributes are listed in Table 1. The approximate number of tokens per control dataset is computed assuming the tokenization mechanism described in Section 2.1.

### 3.1 Simulated control tasks

Our control tasks consist of datasets generated by specialist SoTA or near-SoTA reinforcement learning agents trained on a variety of different environments. For each environment we record a subset of the experience the agent generates (states, actions, and rewards) while it is training.

The simulated environments include Meta-World (Yu et al., 2020) introduced to benchmark meta-reinforcement learning and multi-task learning, Sokoban (Racanière et al., 2017) proposed as a planning problem, BabyAI (Chevalier-Boisvert et al., 2018) for language instruction following in grid-worlds, the DM Control Suite (Tunyasuvunakool et al., 2020) for continuous control, as well as DM Lab (Beattie et al., 2016) designed to teach agents navigation and 3D vision from raw pixels with an egocentric viewpoint. We also use the Arcade Learning Environment (Bellemare et al., 2013) with classic Atari games (we use two sets of

Table 1: **Datasets.** Left: Control datasets used to train Gato. Right: Vision & language datasets. Sample weight means the proportion of each dataset, on average, in the training sequence batches.

| Control environment | Tasks | Episodes | Approx. Tokens | Sample Weight |
|---|---|---|---|---|
| DM Lab | 254 | 16.4M | 194B | 9.35% |
| ALE Atari | 51 | 63.4K | 1.26B | 9.5% |
| ALE Atari Extended | 28 | 28.4K | 565M | 10.0% |
| Sokoban | 1 | 27.2K | 298M | 1.33% |
| BabyAI | 46 | 4.61M | 22.8B | 9.06% |
| DM Control Suite | 30 | 395K | 22.5B | 4.62% |
| DM Control Suite Pixels | 28 | 485K | 35.5B | 7.07% |
| DM Control Suite Random Small | 26 | 10.6M | 313B | 3.04% |
| DM Control Suite Random Large | 26 | 26.1M | 791B | 3.04% |
| Meta-World | 45 | 94.6K | 3.39B | 8.96% |
| Procgen Benchmark | 16 | 1.6M | 4.46B | 5.34% |
| RGB Stacking simulator | 1 | 387K | 24.4B | 1.33% |
| RGB Stacking real robot | 1 | 15.7K | 980M | 1.33% |
| Modular RL | 38 | 843K | 69.6B | 8.23% |
| DM Manipulation Playground | 4 | 286K | 6.58B | 1.68% |
| Playroom | 1 | 829K | 118B | 1.33% |
| Total | 596 | 63M | 1.5T | 85.3% |

| Vision / language dataset | Sample Weight |
|---|---|
| MassiveText | 6.7% |
| M3W | 4% |
| ALIGN | 0.67% |
| MS-COCO Captions | 0.67% |
| Conceptual Captions | 0.67% |
| LTIP | 0.67% |
| OKVQA | 0.67% |
| VQAV2 | 0.67% |
| Total | 14.7% |

games that we call ALE Atari and ALE Atari Extended, see Section F.1 for details). We as well include the Procgen Benchmark (Cobbe et al., 2020) and Modular RL (Huang et al., 2020). We also include four tasks using a simulated Kinova Jaco arm from DM Manipulation Playground, as introduced in Zolna et al. (2020). Section F includes a more in-depth description of these control tasks, along with what RL agent was used to generate the data.

We found it effective to train on a filtered set of episodes with returns at least 80% of the expert return for the task. The expert return measures the maximum sustained performance that the expert agent can achieve. We define it as the maximum over the set of all windowed average returns calculated over all the collected episodes for a task:

$$\max_{j \in [0,1,...,N-W]} \left( \sum_{i=j}^{j+L-1} \frac{R_i}{W} \right)$$

where $N$ it the total number of collected episodes for the task, $W$ is the window size, and $R_i$ is the total return for episode $i$. To obtain accurate estimates, in practice, we set $W$ to be 10% of the total data amount or a minimum of 1000 episodes (i.e. $W = \min(1000, 0.1 \times N)$).

## 3.2 Vision and language

Gato is trained on MassiveText (Rae et al., 2021), a collection of large English-language text datasets from multiple sources: web pages, books, news articles, and code.

We also included several vision-language datasets in Gato's training. ALIGN (Jia et al., 2021) consists of 1.8B images and their alternative text (alt-text) annotations. LTIP (Long Text & Image Pairs), consists of 312 million images with captions (Alayrac et al., 2022). Conceptual captions (Sharma et al., 2018) and COCO captions (Chen et al., 2015) are captioning datasets with 3.3M and 120k image-text pairs respectively. The MultiModal MassiveWeb (M3W) dataset (Alayrac et al., 2022) includes 43M webpages where both text and images were extracted. We also included visual question-answering datasets. In particular OKVQA (Marino et al., 2019) and VQAv2 (Antol et al., 2015) with 9K and 443K triplets of images, questions, and answers. To form a training episode from these, we sample five (image, text) pairs, tokenize them, concatenate, and then pad or randomly crop to the required training sequence length.

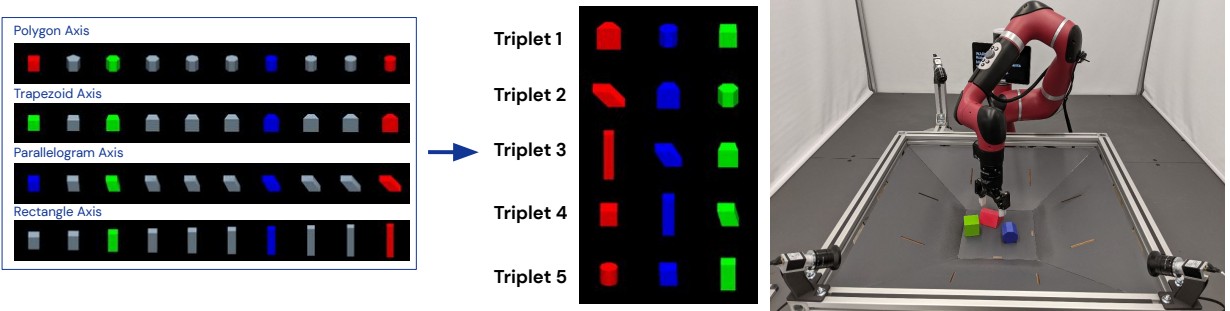

Figure 4: **RGB Stacking environment with the Sawyer robot arm.** Blocks vary along several shape axes, with 5 held out test triplets. The goal is to stack red on blue, ignoring green.

### 3.3 Robotics - RGB Stacking Benchmark (real and sim)

As a testbed for taking physical actions in the real world, we chose the robotic block stacking environment introduced by Lee et al. (2021). The environment consists of a Sawyer robot arm with 3-DoF cartesian velocity control, an additional DoF for velocity, and a discrete gripper action. The robot's workspace contains three plastic blocks colored red, green and blue with varying shapes. The available observations include two $128 \times 128$ camera images, robot arm and gripper joint angles as well as the robot's end-effector pose. Notably, ground truth state information for the three objects in the basket is not observed by the agent. Episodes have a fixed length of 400 timesteps at 20 Hz for a total of 20 seconds, and at the end of an episode block positions are randomly re-positioned within the workspace. The robot in action is shown in Figure 4. There are two challenges in this benchmark: *Skill Mastery* (where the agent is provided data from the 5 test object triplets it is later tested on) and *Skill Generalization* (where data can only be obtained from a set of training objects that excludes the 5 test sets).

We used several sources of training data for these tasks. In Skill Generalization, for both simulation and real, we use data collected by the best generalist sim2real agent from Lee et al. (2021). We collected data only when interacting with the designated RGB-stacking *training objects* (this amounts to a total of 387k successful trajectories in simulation and 15k trajectories in real). For Skill Mastery we used data from the best per group experts from Lee et al. (2021) in simulation and from the best sim2real policy on the real robot (amounting to 219k trajectories in total). Note that this data is only included for specific Skill Mastery experiments in Section 5.4.

## 4 Capabilities of the generalist agent

In this section, we summarize the performance of Gato when trained on the above described data. That is, all results across all tasks are derived from a single pretrained model with a single set of weights. Results with fine-tuning will be presented in Section 5.

### 4.1 Simulated control tasks

Figure 5 shows the number of distinct control tasks for which Gato performs above a given score threshold, relative to expert performance demonstrated in Gato's training data.

We report performance as a percentage, where 100% corresponds to the per-task expert and 0% to a random policy. For each simulated control task we trained our model on, we roll out the Gato policy on the corresponding environment 50 times and average the defined scores. As shown in Figure 5, Gato performs over 450 out of 604 tasks at over a 50% expert score threshold.

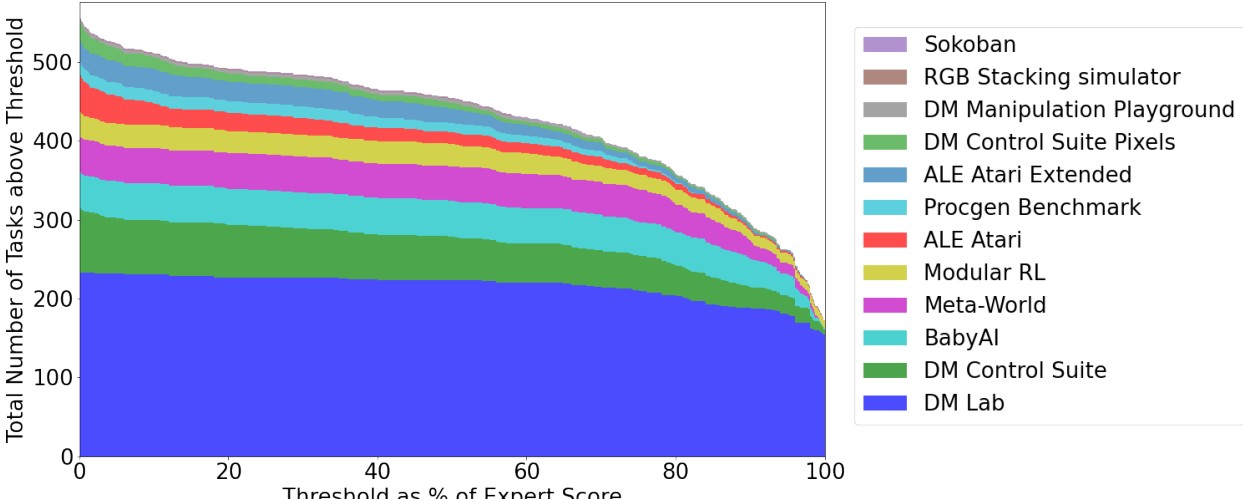

Figure 5: **Gato's performance on simulated control tasks.** Number of tasks where the performance of the pretrained model is above a percentage of expert score, grouped by domain. Here values on the x-axis represent a specific percentage of expert score, where 0 corresponds to random agent performance. The y-axis is the number of tasks where the pretrained model's mean performance is equal to or above that percentage. That is, the width of each colour band indicates the number of tasks where Gato's mean performance is above a percentage of the maximum score obtained by a task-specific expert.

In ALE Atari (Bellemare et al., 2013) Gato achieves the average human (or better) scores for 23 Atari games[1], achieving over twice human score for 11 games. While the single-task online RL agents which generated the data still outperform Gato, this may be overcome by adding capacity or using offline RL training rather than purely supervised (see Section 5.5 where we present a specialist single domain ALE Atari agent achieving better than human scores for 44 games).

On BabyAI (Chevalier-Boisvert et al., 2018) Gato achieves over 80% of expert score for nearly all levels[2]. For the most difficult task, called BossLevel, Gato scores 75%. The two other published baselines we could find, BabyAI 1.0 and BabyAI 1.1 (Hui et al., 2020), scored 77% and 90%, respectively, having trained on this single task alone using a million demonstrations.

On Meta-World (Yu et al., 2020) Gato achieves more than 50% for all 44 out of 45 tasks that we trained on, over 80% for 35 tasks, and over 90% for 3 tasks. On canonical DM Control Suite (Tassa et al., 2018), Gato achieves better than 50% of the expert score on 21 out of 30 tasks from state, and more than 80% for 18 tasks.

## 4.2 Robotics

First person teleoperation enables the collection of expert demonstrations. However, such demonstrations are slow and costly to collect. Data-efficient behavior cloning methods are therefore desirable for training a generalist robot manipulator and offline pretraining is thus a well-motivated area of research. To that end, we evaluated Gato on the established RGB Stacking benchmark for robotics.

---

[1]The full list of games: Assault, Atlantis, Bank heist, Battle zone, Bowling, Crazy climber, Defender, Fishing derby, Gopher, Hero, Ice hockey, Jamesbond, Kangaroo, Kung fu master, Name this game, Pong, Road runner, Robotank, Tennis, Time pilot, Up n down, Wizard of wor, Zaxxon.

[2]The only three tasks below 80% success rate are GoToImpUnlock (59%), Unlock (74%), and BossLevel (75%).

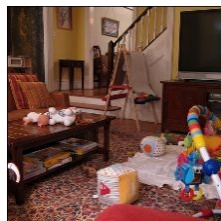

The colorful ceramic toys are on the living room floor.

a living room with three different color deposits on the floor

a room with a long red rug a tv and some pictures

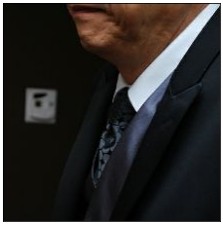

Man standing in the street wearing a suit and tie.

A man in a blue suit with a white bow tie and black shoes.

A man with a hat in his hand looking at the camera

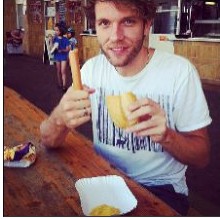

A bearded man is holding a plate of food.

Man holding up a banana to take a picture of it.

a man smiles while holding up a slice of cake

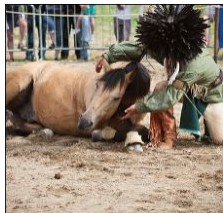

a group of people that is next to a big horse

A tan horse holding a piece of cloth lying on the ground.

Two horses are laying on their side on the dirt.

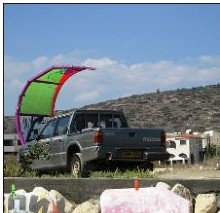

Man biting a kite while standing on a construction site

a big truck in the middle of a road

A truck with a kite painted on the back is parked by rocks.

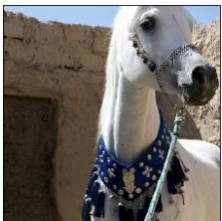

a white horse with a blue and silver bridle

A white horse with blue and gold chains.

A horse is being shown behind a wall.

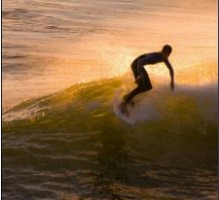

a couple of people are out in the ocean

A surfer riding a wave in the ocean.

A surfer with a wet suit riding a wave.

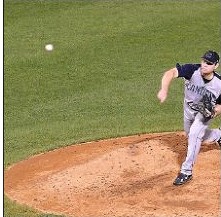

A baseball player pitching a ball on top of a baseball field.

A man throwing a baseball at a pitcher on a baseball field.

A baseball player at bat and a catcher in the dirt during a baseball game

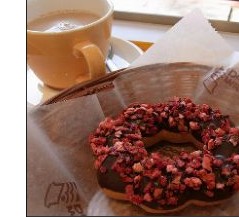

Pistachios on top of a bowl with coffee on the side.

A bowl and a glass of liquid sits on a table.

A white plate filled with a banana bread next to a cup of coffee.

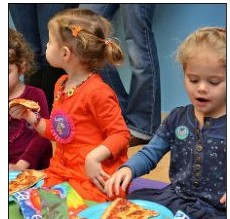

A group of children eating pizza at a table.

Two boys having pizza for lunch with their friends.

The boys are eating pizza together at the table.

Figure 6: **Image captions generated by Gato.** Gato prompted to be an image captioner, describing the first several held-out images from MS-COCO. We report the first three captions sampled using temperature 0.9, without cherry-picking. The prompt is shown in the appendix.

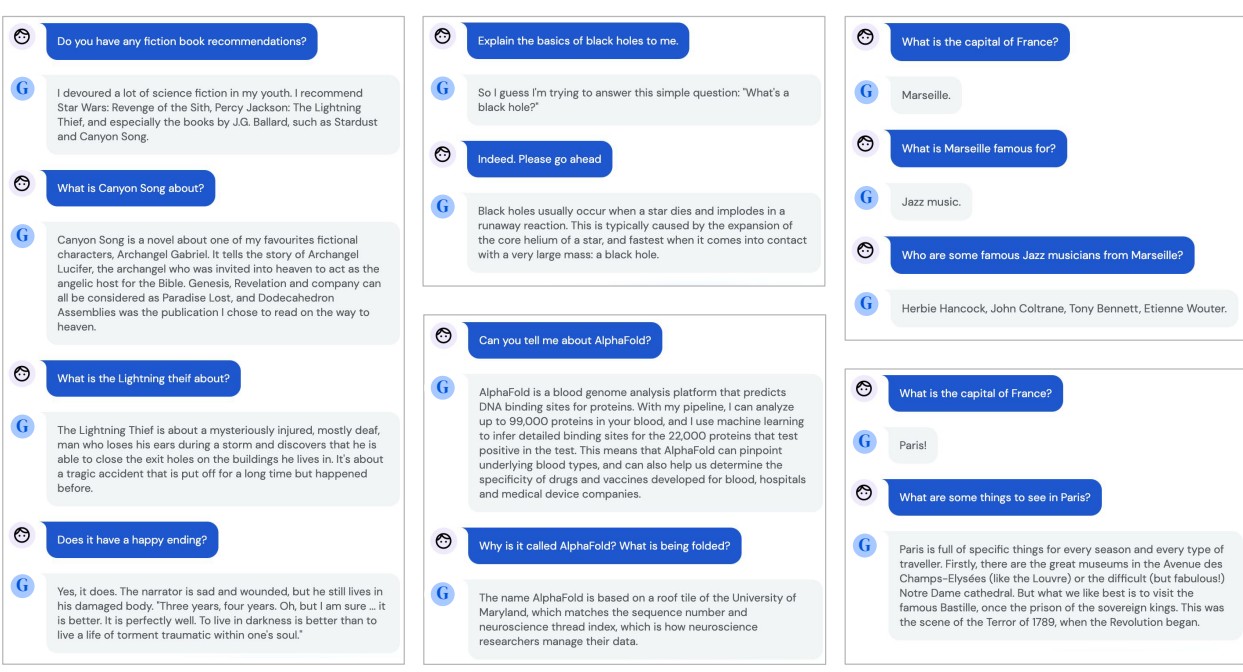

Figure 7: **Chitchat with Gato.** Dialogues with Gato when it is prompted to be a chat bot. Usually Gato replies with a relevant response, but is often superficial or factually incorrect, which could likely be improved with further scaling. We used the same prompt as in Rae et al. (2021).

Table 2: **Gato real robot Skill Generalization results.** In addition to performing hundreds of other tasks, Gato also stacks competitively with the comparable published baseline.

| Agent | Group 1 | Group 2 | Group 3 | Group 4 | Group 5 | Average |
|---|---|---|---|---|---|---|
| Gato | **24.5**% | 33% | **50.5**% | 76.5% | **66.5**% | **50.2**% |
| BC-IMP (Lee et al., 2021) | 23% | **39.3**% | 39.3% | **77.5**% | 66% | 49% |

**Skill Generalization Performance**

The Skill Generalization challenge from the RGB Stacking robotics benchmark tests the agent's ability to stack objects of previously unseen shapes. The agent is trained on a dataset consisting of episodes of the robot stacking objects with a variety of different shapes. Five triplets of object shapes are, however, not included in the training data and serve as test triplets. We evaluated the trained generalist for 200 episodes per test triplet on the real robot. Table 2 shows that our generalist agent's success rate on each test triplet is comparable to the single task BC-IMP (filtered BC) baseline in Lee et al. (2021).

### 4.3 Text samples

The model demonstrates rudimentary dialogue and image captioning capabilities. Figure 6 contains a representative sample of Gato's image captioning performance. Figure 7 shows some hand-picked examples of plain text dialogue exchange.

## 5 Analysis

### 5.1 Scaling Laws Analysis

In Figure 8, we analyze the aggregate in-distribution performance of the pretrained model as a function of the number of parameters in order to get insight into how performance could improve with increased model capacity. We evaluated 3 different model sizes (measured in parameter count): a `79M model`, a `364M model`, and a `1.18B model` (Gato). We refer to Section C for details on the three model architectures.

Here, for all three model sizes we plot the normalized return as training progresses. To get this single value, for each task we calculate the performance of the model as a percentage of expert score (the same as done in Section 4.1). Then for each domain listed in Table 1 we average the percentage scores across all tasks for that domain. Finally, we mean-aggregate the percentage scores across all domains. We can see that for an equivalent token count, there is a significant performance improvement with increased scale.

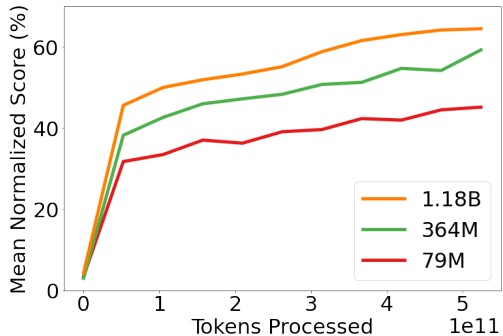

Figure 8: **Model size scaling laws results.** In-distribution performance as a function of tokens processed for 3 model scales. Performance is first mean-aggregated within each separate control domain, and then mean-aggregated across all domains. We can see a consistent improvement as model capacity is increased for a fixed number of tokens.

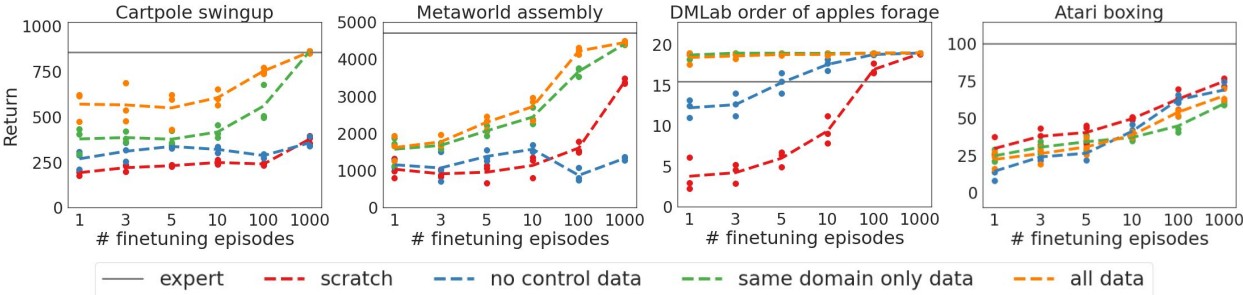

Figure 9: **Few-shot performance, ablating over various pretraining settings.** Orange corresponds to the base Gato pretrained on all data. Red is trained from scratch only on the few-shot data. `364M` parameter variants of Gato were used for this experiment to save compute.

## 5.2  Out of distribution tasks

In this section we want to answer the following question: *Can our agent be used to solve a completely new task efficiently?* For this reason, we held-out all data for four tasks from our pre-training set: `cartpole.swingup` (DM Control Suite domain), `assembly-v2` (Meta-World domain), `order_of_apples_forage_simple` (DM Lab domain), and `boxing` (ALE Atari domain). These four tasks will serve as testbeds for evaluating the out-of-distribution capabilities of Gato.

Ideally, the agent could potentially learn to adapt to a new task via conditioning on a prompt including demonstrations of desired behaviour. However, due to accelerator memory constraints and the extremely long sequence lengths of tokenized demonstrations, the maximum context length possible does not allow the agent to attend over an informative-enough context. Therefore, to adapt the agent to new tasks or behaviours, we choose to fine-tune the agent's parameters on a limited number of demonstrations of a single task, and then evaluate the fine-tuned model's performance in the environment. Fine-tuning is very similar to pretraining with minor changes, such as different learning rate schedule; see Section E for details.

We want to measure how choice of data used during pretraining influences post-fine-tuning performance. To this end, we compare Gato (trained on *all data*) to variants trained on ablated datasets:

1. A model pretrained only on data from the same domain as the task to be fine-tuned on, *same domain only data.*

2. A model pretrained only on non-control data, *no control data.*

3. A model fine-tuned from scratch, i.e. no pretraining at all, *scratch.*

Considering as all these experiments require training a new model from scratch and then also fine-tuning, we present results using the less compute-intensive `364M` parameter architecture described in Section 5.1. Results are shown in Figure 9.

Fine-tuning performance on both `cartpole.swingup` and `assembly-v2` tasks, both of which do not require image processing, present similar trends. Pretraining on all the datasets yields the best results, followed by pretraining on the same domain only. This difference is smaller for `assembly-v2` but consistent for all few shot datasets. For these non-image-based environments, we see either no benefit (`cartpole.swingup`) or even negative transfer (`assembly-v2`) when pretraining on *no control* datasets, which only contain images and text data.

Results for DM Lab `order_of_apples_forage_simple` are slightly different. Pretraining on DM Lab data only is already enough to approach the maximum reward of 19 and hence there is no observable benefit of adding data from different environments. What is different when compared to previously analysed no-vision environments is that pretraining on *no control* data helps, which can be possibly explained by the fact that

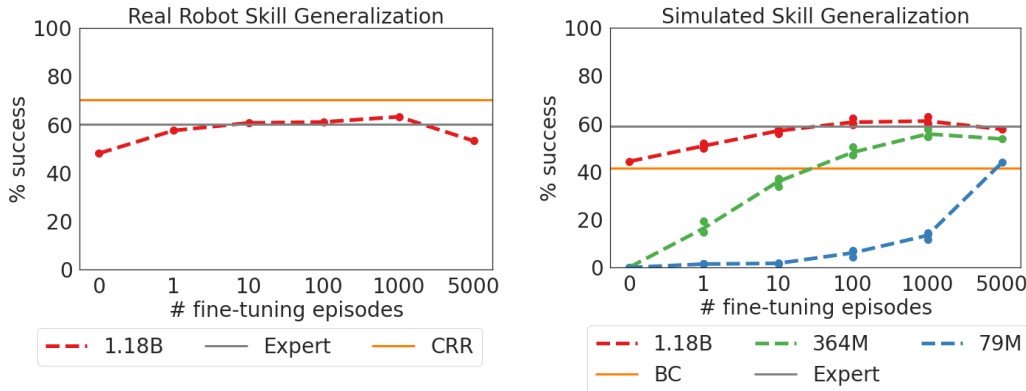

Figure 10: **Robotics fine-tuning results.** Left: Comparison of real robot Skill Generalization success rate averaged across test triplets for Gato, expert, and CRR trained on 35k expert episodes (upper bound). Right: Comparison of simulated robot Skill Generalization success rate averaged across test triplets for a series of ablations on the number of parameters, including scores for expert and a BC baseline trained on 5k episodes.

agents in the DM Lab environment are fed images which, despite being simulated, are natural looking. Therefore, transfer from image captioning or visual grounded question answering tasks is possible.

We were not able to observe any benefit from pretraining on `boxing`. The randomly initialized model seems to work better than any of the pretrained variants considered. We hypothesise that this is caused by the game's input images being visually very distinct from the other data, suggesting transfer is difficult. We discuss this Atari challenge further in our related work section.

### 5.3 Fine-tuning on Robotic Stacking Tasks

Section 4.2 demonstrates that the base Gato capable of a diverse array of tasks can perform competitively on the RGB Stacking Skill Generalization benchmark. In this section, we would like to answer the following question: *How does our agent improve on robotics tasks when allowed to fine-tune similarly to how we fine-tune on new tasks in Section 5.2?* We consider different model sizes and analyse the impact of pretraining datasets on the Skill Generalization benchmark, as well as a novel out of distribution task. Further analysis of fine-tuning with dataset ablations is in Appendix I.

**Skill Generalization**

First, we would like to show that fine-tuning on object-specific data, similarly to what was done by Lee et al. (2022), is beneficial. Therefore, we fine-tuned Gato separately on five subsets of demonstrations from the *test* dataset. Each subset was obtained by random partitioning of a test dataset consisting of demonstrations gathered by a generalist sim-to-real agent stacking real test objects. We consider this setting, which is comparable to the fine-tuning baselines on RGB stacking tasks from (Lee et al., 2022); and use the 5k dataset that their behavior cloning 5k results are obtained with. To best match their experiments, we change our return filtering scheme during training: instead of using only successful stacks, we condition on the normalized return of the episode.

Figure 10 compares the success rate of Gato across different fine-tuning data regimes to the sim-to-real expert and a Critic-Regularized Regression (CRR) (Wang et al., 2020) agent trained on 35k episodes of all test triplets. Gato, in both reality and simulation (red curves on the left and right figure, respectively), recovers the expert's performance with only 10 episodes, and peaks at 100 or 1000 episodes of fine-tuning data, where it exceeds the expert. After this point (at 5000), performance degrades slightly but does not drop far below the expert's performance.

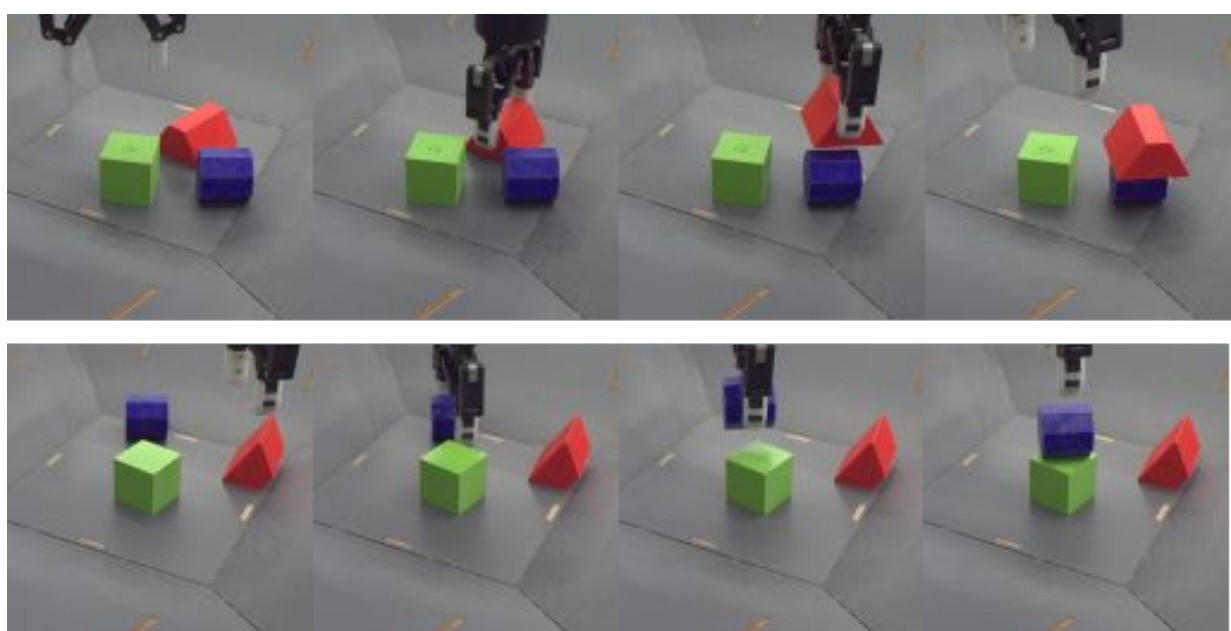

Figure 11: **Comparing training/test task goal variations.** Top: the standard "stack red on blue" task tested in the Skill Generalization benchmark. Bottom: the novel "stack blue on green" task demonstrating Gato's out of distribution adaptation to perceptual variations.

### Fine-tuning and Model Size

To better understand the benefit of large models for few-shot adaptation in robotics domains, we conducted an ablation on model parameter size. This section focuses on in-simulation evaluation. Figure 10 compares the full `1.18B` parameter Gato with the smaller `364M` and `79M` parameter variants for varying amounts of fine-tuning data. Although the `364M` model overfits on one episode, causing performance to drop, there is a clear trend towards better adaptation with fewer episodes as the number of parameters is scaled up. The `79M` model performs clearly worse than its bigger counterparts. The results suggest that the model's greater capacity allows the model to use representations learned from the diverse training data at test time.

### Adaptation to Perceptual Variations

While the Skill Generalization task is an effective benchmark for motor Skill Generalization to shape variations, it does not test the agent's ability to adapt to perceptual variations and permutations in the objective specification. To further evaluate Gato's generalization capabilities, we devised a new task in the RGB stacking benchmark where the goal is to stack the blue object on the green object, for test triplet 1 (see Figure 11). First, we used a 3D mouse to collect 500 demonstrations of this task on the real robot, for a total of 2 hours and 45 minutes of demonstration data, and fine-tuned Gato on these episodes. Notably, all of the simulated and real robotics data in the pretraining set shows the robot successfully stacking the red object on the blue object, and the data does not include the object shapes in the test set. We found that additionally adding simulated demonstrations of the stack blue on green task to the fine-tuning dataset improved performance, and 10% was an ideal sampling ratio for this data.

We achieved a final 60% success rate after evaluating fine-tuned Gato on the real robot, while a BC baseline trained from scratch on the blue-on-green data achieved only 0.5% success (1/200 episodes). Qualitatively, the BC baseline would consistently move towards the blue object and occasionally pick it up and place it on top of the green object, but a full, stable stack was almost never achieved.

Table 3: **Real robot Skill Mastery results.** Gato is competitive with the filtered BC baseline.

| AGENT | GROUP 1 | GROUP 2 | GROUP 3 | GROUP 4 | GROUP 5 | AVERAGE |
|---|---|---|---|---|---|---|
| GATO | 58% | 57.6% | **78.5%** | **89** % | **95.1%** | **75.6%** |
| BC-IMP (LEE ET AL., 2021) | **75.6%** | **60.8%** | 70.8% | 87.8% | 78.3% | 74.6% |

### 5.4 Robotics: Skill Mastery

Similarly to the Skill Generalization challenge discussed in Section 4.2, the Skill Mastery challenge consists in training a robotic arm to stack blocks of different shapes. However, the Skill Mastery allows the agent to train on data involving the object shapes used for evaluation, i.e. the *test* set in Skill Generalization becomes a part of the Skill Mastery *training* set. Thus, this challenge serves to measure Gato's performance on in-distribution tasks (possibly with initial conditions not seen in the training demonstrations). Our Skill Mastery results use an earlier version of the Gato architecture described in Appendix H, with no fine-tuning.

Table 3 compares the group-wise success percentage and the average success across object groups for Gato and the established BC-IMP baseline. Gato exceeds or closely matches BC-IMP's performance on all but one training triplet.

### 5.5 Specialist single-domain multi-task agents

In this section we show results obtained with two specialist (rather than generalist) agents. Both of them were trained on data from a single domain only and rolled out 500 times for each training task without any per-task fine-tuning.

#### Meta-World

The first agent uses the smallest architecture introduced in Section 5.1, i.e. `79M` parameters, and is trained on all 50 Meta-World tasks. While Gato has access to the state of the MuJoCo physics engine and unlimited task seeds, the agent presented here has no access to any extra features or tasks and uses the canonical API as in (Yu et al., 2020). This experiment is to show that the architecture proposed in our paper can be used to obtain state-of-the-art agents also at small scale. The training procedure was to train single-task MPO (Abdolmaleki et al., 2018) experts on each of the MT-50 tasks individually, recording the trajectories produced while training. This experience is then combined, or distilled, into a single agent, which achieves 96.6% success rate averaged over all 50 tasks. To the best of our knowledge this agent is the first one to accomplish nearly 100% average success rate simultaneously (multi-task) for this benchmark. See Table 7 in the supplementary material (Section K) for the full list of tasks and corresponding success rates of our agent.

#### ALE Atari

We also trained a specialist agent on all 51 ALE Atari tasks. As the Atari domain is much more challenging than Meta-World, we used the Gato architecture with `1.18B` parameters.

The resulting agent performs better than the average human for 44 games (see Section 4.1 for details on our evaluation and scoring). We want to note that the performance of online experts used to generate training data for the other 7 games were also below the average human. Hence, the specialist Atari agent achieved better than human performance for all games where data contained super-human episodes.

The specialist Atari agent outperforms our generalist agent Gato, which achieved super-human performance on 23 games. It suggests that scaling Gato may result in even better performance. We, however, purposely restricted Gato's size such that it can be run in real-time on the real robot.

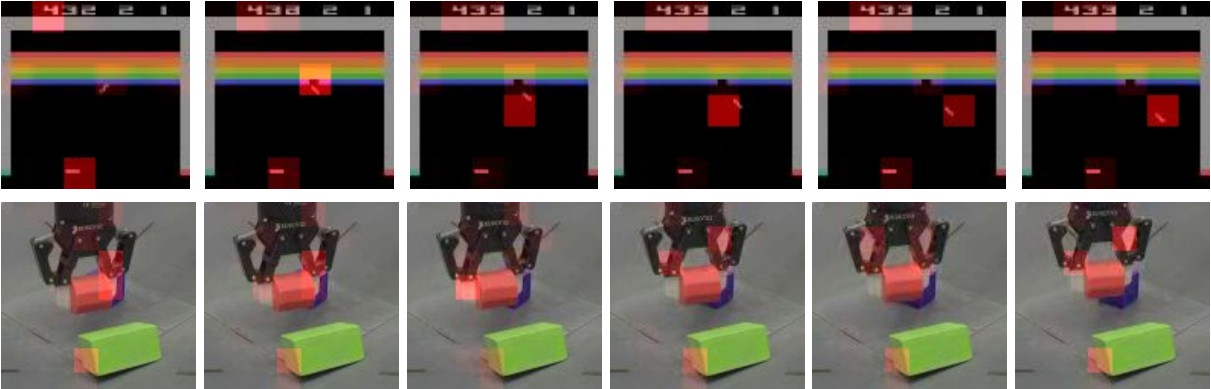

Figure 12: **Attention maps.** Time-lapse attention maps from selected heads at the first layer for Atari Breakout and RGB Stacking.

## 5.6 Attention Analysis

We rendered the transformer attention weights over the image observations for various tasks, to gain a qualitative sense of how Gato attends to different regions of the image across tasks (see Figure 12). Further details and visualizations for more tasks can be found in Appendix J. These visualizations clearly show that attention tracks the task-relevant objects and regions.

## 5.7 Embedding Visualization

To understand how Gato encodes differently information per task, we visualized per-task embeddings.

We analysed 11 tasks. For each task, we randomly sample 100 episodes and tokenize each of them. Then, from each episode we take a subsequence of 128 tokens, compute their embeddings (at layer 12, which is half the total depth of the transformer layers) and average them over the sequence. The averaged embeddings for all tasks are used as input to PCA, which reduces their dimensionality to 50. Then, T-SNE is used to get the final 2D embeddings.

Figure 13 shows the final T-SNE embeddings plotted in 2D, colorized by task. Embeddings from the same tasks are clearly clustered together, and task clusters from the same domain and modality are also located close to each other. Even held-out task (`cartpole.swingup`) is clustered correctly and lays next to another task from DM Control Suite Pixels.

## 6 Related Work

The most closely related architectures to that of Gato are Decision Transformers (Chen et al., 2021b; Reid et al., 2022; Zheng et al., 2022; Furuta et al., 2021) and Trajectory Transformer (Janner et al., 2021), which showed the usefulness of highly generic LM-like architectures for a variety of control problems. Gato also uses an LM-like architecture for control, but with design differences chosen to support multi-modality, multi-embodiment, large scale and general purpose deployment. Pix2Seq (Chen et al., 2022) also uses an LM-based architecture for object detection. Perceiver IO (Jaegle et al., 2021) uses a transformer-derived architecture specialized for very long sequences, to model any modality as a sequence of bytes. This and similar architectures could be used to expand the range of modalities supported by future generalist models.

Gato was inspired by works such as GPT-3 (Brown et al., 2020) and Gopher (Rae et al., 2021), pushing the limits of generalist language models; and more recently the Flamingo (Alayrac et al., 2022) generalist visual language model. Chowdhery et al. (2022) developed the 540B parameter Pathways Language Model (PalM)

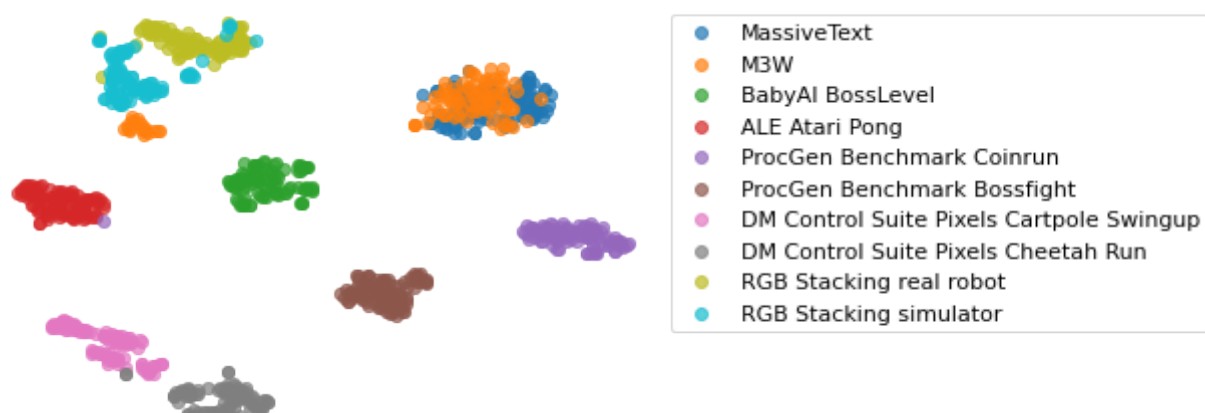

Figure 13: **Embedding visualization.** T-SNE visualization of embeddings from different tasks. A large part of the vision-language embeddings (M3W) overlaps with the language cluster (MassiveText). Other tasks involving actions fall in their own cluster.

explicitly as a generalist few-shot learner for hundreds of text tasks. Future work should consider how to unify these text capabilities into one fully generalist agent that can also act in real time in the real world, in diverse environments and embodiments.

Gato also takes inspiration from recent works on multi-embodiment continuous control. Huang et al. (2020) used message passing graph networks to build a single locomotor controller for many simulated 2D walker variants. Kurin et al. (2020) showed that transformers can outperform graph based approaches for incompatible (i.e. varying embodiment) control, despite not encoding any morphological inductive biases. Devin et al. (2017) learn a modular policy for multi-task and multi-robot transfer in simulated 2D manipulation environments. Chen et al. (2018) train a universal policy conditioned on a vector representation of robot hardware, showing successful transfer both to simulated held out robot arms, and to a real world sawyer robot arm.

A variety of earlier generalist models have been developed that, like Gato, operate across highly distinct domains and modalities. NPI (Reed & De Freitas, 2016) trained a single LSTM (Hochreiter & Schmidhuber, 1997) to execute diverse programs such as sorting an array and adding two numbers, such that the network is able to generalize to larger problem instances than those seen during training. Kaiser et al. (2017) developed the MultiModel that trains jointly on 8 distinct speech, image and text processing tasks including classification, image captioning and translation. Modality-specific encoders were used to process text, images, audio and categorical data, while the rest of the network parameters are shared across tasks. Schmidhuber (2018) proposed "*one big net for everything*", describing a method for the incremental training of an increasingly general problem solver. Keskar et al. (2019) proposed controllable multi-task language models that can be directed according to language domain, subdomain, entities, relationships between entities, dates, and task-specific behavior.

In this discussion, it is important to distinguish between one single multi-task network architecture versus one single neural network with the same weights for all tasks. Several poplar RL agents achieve good multi-task RL results within single domains such as Atari57 and DMLab (Espeholt et al., 2018; Song et al., 2020; Hessel et al., 2019). However, it is much more common to use the same policy architecture and hyper-parameters across tasks, but the policy parameters are different in each task (Mnih et al., 2015; Tassa et al., 2018). This is also true of state-of-the-art RL methods applied to board games (Schrittwieser et al., 2020). Moreover, this choice has been adopted by off-line RL benchmarks (Gulcehre et al., 2020; Fu et al., 2020) and recent works on large sequence neural networks for control, including decision transformers (Chen et al., 2021b; Reid et al., 2022; Zheng et al., 2022) and the Trajectory Transformer of Janner et al. (2021). In contrast, in this work we learn a single network with the same weights across a diverse set of tasks.

Recent position papers advocate for highly generalist models, notably Schmidhuber (2018) proposing one big net for everything, and Bommasani et al. (2021) on foundation models. However, to our knowledge there has not yet been reported a single generalist trained on hundreds of vision, language and control tasks using modern transformer networks at scale.

"Single-brain"-style models have interesting connections to neuroscience. Mountcastle (1978) famously stated that "*the processing function of neocortical modules is qualitatively similar in all neocortical regions. Put shortly, there is nothing intrinsically motor about the motor cortex, nor sensory about the sensory cortex*". Mountcastle found that columns of neurons in the cortex behave similarly whether associated with vision, hearing or motor control. This has motivated arguments that we may only need one algorithm or model to build intelligence (Hawkins & Blakeslee, 2004).

Sensory substitution provides another argument for a single model (Bach-y Rita & Kercel, 2003). For example, it is possible to build tactile visual aids for blind people as follows. The signal captured by a camera can be sent via an electrode array on the tongue to the brain. The visual cortex learns to process and interpret these tactile signals, endowing the person with some form of "vision". Suggesting that, no matter the type of input signal, the same network can process it to useful effect.

Our work is based on deep autoregressive models, which have a long history and can be found in generative models of text, images, video and audio. Combining autoregressive generation with transformers (Vaswani et al., 2017; Devlin et al., 2018) has been of enormous impact in language modelling (Brown et al., 2020; Rae et al., 2021), protein folding (Jumper et al., 2021), vision-language models (Tsimpoukelli et al., 2021; Wang et al., 2021; Alayrac et al., 2022), code generation (Chen et al., 2021c; Li et al., 2022b), dialogue systems with retrieval capabilities (Nakano et al., 2021; Thoppilan et al., 2022), speech recognition (Pratap et al., 2020), neural machine translation (Johnson et al., 2019) and more (Bommasani et al., 2021). Recently researchers have explored task decomposition and grounding with language models (Huang et al., 2022; Ahn et al., 2022).

Li et al. (2022a) construct a control architecture, consisting of a sequence tokenizer, a pretrained language model and a task-specific feed-forward network. They apply it to VirtualHome and BabyAI tasks, and find that the inclusion of the pretrained language model improves generalisation to novel tasks. Similarly, Parisi et al. (2022) demonstrate that vision models pretrained with self-supervised learning, especially crop segmentations and momentum contrast (He et al., 2020), can be effectively incorporated into control policies.

As mentioned earlier, transfer in Atari is challenging. Rusu et al. (2016) researched transfer between randomly selected Atari games. They found that Atari is a difficult domain for transfer because of pronounced differences in the visuals, controls and strategy among the different games. Further difficulties that arise when applying behaviour cloning to video games like Atari are discussed by Kanervisto et al. (2020).

There has been great recent interest in data-driven robotics (Cabi et al., 2019; Chen et al., 2021a). However, Bommasani et al. (2021) note that in robotics "*the key stumbling block is collecting the right data. Unlike language and vision data, robotics data is neither plentiful nor representative of a sufficiently diverse array of embodiments, tasks, and environments*". Moreover, every time we update the hardware in a robotics lab, we need to collect new data and retrain. We argue that this is precisely why we need a generalist agent that can adapt to new embodiments and learn new tasks with few data.

Generating actions using an autoregressive model can lead to causal "self-delusion" biases when there are confounding variables (Ortega et al., 2021). For example, sampling actions can condition the model to solve the wrong task when multiple tasks share similar observation and actions specifications. As explained in Section 2, we use prompt engineering in ambiguous tasks, conditioning our model on a successful demonstration. This screens off confounding variables, reducing self-delusions. Another solution which we did not explore in this work is to use counterfactual teaching, where we train a model online using instantaneous expert feedback. We leave this for future investigation.

# 7   Broader Impact

Although generalist agents are still only an emerging area of research, their potential impact on society calls for a thorough interdisciplinary analysis of their risks and benefits. For the sake of transparency, we document the intended use cases of Gato in the model card in Appendix A. However, the tools for mitigating harms of generalist agents are relatively underdeveloped, and require further research before these agents are deployed.

Since our generalist agent can act as a vision-language model, it inherits similar concerns as discussed in (Weidinger et al., 2021; Bommasani et al., 2021; Rae et al., 2021; Alayrac et al., 2022). In addition, generalist agents can take actions in the the physical world; posing new challenges that may require novel mitigation strategies. For example, physical embodiment could lead to users anthropomorphizing the agent, leading to misplaced trust in the case of a malfunctioning system, or be exploitable by bad actors. Additionally, while cross-domain knowledge transfer is often a goal in ML research, it could create unexpected and undesired outcomes if certain behaviors (e.g. arcade game fighting) are transferred to the wrong context. The ethics and safety considerations of knowledge transfer may require substantial new research as generalist systems advance.

Technical AGI safety (Bostrom, 2017) may also become more challenging when considering generalist agents that operate in many embodiments. For this reason, preference learning, uncertainty modeling and value alignment (Russell, 2019) are especially important for the design of human-compatible generalist agents. It may be possible to extend some of the value alignment approaches for language (Ouyang et al., 2022; Kenton et al., 2021) to generalist agents. However, even as technical solutions are developed for value alignment, generalist systems could still have negative societal impacts even with the intervention of well-intentioned designers, due to unforeseen circumstances or limited oversight (Amodei et al., 2016). This limitation underscores the need for a careful design and a deployment process that incorporates multiple disciplines and viewpoints.

Understanding how the models process information, and any emergent capabilities, requires significant experimentation. External retrieval (Borgeaud et al., 2021; Menick et al., 2022; Nakano et al., 2021; Thoppilan et al., 2022) has been shown to improve both interpretability and performance, and hence should be considered in future designs of generalist agents.

Although still at the proof-of-concept stage, the recent progress in generalist models suggests that safety researchers, ethicists, and most importantly, the general public, should consider their risks and benefits. We are not currently deploying Gato to any users, and so anticipate no immediate societal impact. However, given their potential impact, generalist models should be developed thoughtfully and deployed in a way that promotes the health and vitality of humanity.

# 8   Limitations and Future work

## 8.1   RL data collection

Gato is a data-driven approach, as it is derived from imitation learning. While natural language or image datasets are relatively easy to obtain from the web, a web-scale dataset for control tasks is not currently available. This may seem at first to be problematic, especially when scaling Gato to a higher number of parameters.

That being said, there has already been extensive investigation into this issue. Offline RL aims at leveraging existing control datasets, and its increasing popularity has already resulted in the availability of more diverse and larger datasets. Richer environments and simulations are being built (e.g. Metaverse), and increasing numbers of users already interact with them among thousands of already deployed online games (e.g. there exists a large dataset of Starcraft 2 games). Real-life data has also been already stored for ML research purposes; for example, data for training self-driving cars is acquired from recording human driver data. Finally, while Gato uses data consisting of both observations and corresponding actions, the possibility of using large scale observation-only data to enhance agents has been already studied (Baker et al., 2022).

Thanks to online video sharing and streaming platforms such as Youtube and Twitch, observation-only datasets are not significantly more difficult to collect than natural language datasets, motivating a future research direction to extend Gato to learn from web data.

While the previous paragraph focuses on alleviating drawbacks of data collection from RL agents, it is important to note that this approach presents a different set of tradeoffs compared to scraping web data and can be actually more practical in some situations. Once the simulation is set up and near SOTA agent trained, it can be used to generate massive amounts of high quality data. That is in contrast to the quality of web data which is notorious for its low quality.

In short, we believe that acquiring suitable data is another research question on its own, and this is an active area of research with growing momentum and importance.

### 8.2 Prompt and short context

Gato is prompted with an expert demonstration, which aids the agent to output actions corresponding to the given task. This is particularly useful since there is otherwise no task identifier available to the agent (that is in contrast to many multi-task RL settings). Gato infers the relevant task from the observations and actions in the prompt.

However, the context length of our agent is limited to 1024 tokens which translates to the agent sometimes attending to only a few environment timesteps in total. This is especially the case for environments with image observations, where depending on the resolution each observation can result in more than one hundred tokens each. Hence for certain environments only a short chunk of a demonstration episode fits in the transformer memory.

Due to this limited prompt context, preliminary experiments with different prompt structures resulted in very similar performance. Similarly, early evaluations of the model using prompt-based in-context learning on new environments did not show a significant performance improvement compared to prompt-less evaluation in the same setting.

Context-length is therefore a current limitation of our architecture, mainly due to the quadratic scaling of self-attention. Many recently proposed architectures enable a longer context at greater efficiency and these innovations could potentially improve our agent performance. We hope to explore these architectures in future work.

## 9 Conclusions

Transformer sequence models are effective as multi-task multi-embodiment policies, including for real-world text, vision and robotics tasks. They show promise as well in few-shot out-of-distribution task learning. In the future, such models could be used as a default starting point via prompting or fine-tuning to learn new behaviors, rather than training from scratch.

Given scaling law trends, the performance across all tasks including dialogue will increase with scale in parameters, data and compute. Better hardware and network architectures will allow training bigger models while maintaining real-time robot control capability. By scaling up and iterating on this same basic approach, we can build a useful general-purpose agent.

## Acknowledgments

We would like to thank Dan Horgan, Manuel Kroiss, Mantas Pajarskas, and Thibault Sottiaux for their help with data storage infrastructure; Jean-Baptiste Lespiau and Fan Yang for help on concurrent evaluation; Joel Veness for advising on the model design; Koray Kavukcuoglu for helping inspire the project and facilitating feedback; Tom Erez for advising on the agent design and task selection for continuous control; Igor Babuschkin for helping code the initial prototype; Jack Rae for advising on the transformer language model codebase; Thomas Lampe for building robot infrastructure and advising on real robotics experiments; Boxi Wu for input on ethics and safety considerations; Pedro A. Ortega for advice in regard to causality and self-delusion biases.

## Author Contributions

**Scott Reed** developed the project concept, wrote the initial prototype, and led the project overall.

**Konrad Żołna** led architecture development for vision and text, built infrastructure for tokenization and prompting, and contributed heavily to overall agent development and evaluation.

**Emilio Parisotto** led work on optimizing the transformer architecture, ran the largest number of experiments, and analyzed scaling law properties and in-distribution agent performance.

**Sergio Gómez Colmenarejo** was the technical lead, responsible for creating a scalable data loader and evaluator supporting hundreds of tasks at once, and for the initial robot integration with Gato.

**Alexander Novikov** developed the model including the sampler for the initial prototype, carried out experiments focusing on robotics, and created visualizations.

**Gabriel Barth-Maron** built scalable storage infrastructure to provide Gato with SoTA-level agent experience in Atari and other domains.

**Mai Giménez** conducted large scale agent data collection, built substantial data loading infrastructure, and integrated large scale visual-language datasets into the training of Gato.

**Yury Sulsky** contributed broadly to the Gato codebase including a bespoke distributed training sequence loader, and led the development of benchmarks for out-of-distribution generalization, and the training of competitive baseline agents.

**Jackie Kay** supported physical robotics infrastructure, conducted numerous evaluations and experiments to analyze the generalization properties of Gato, and contemplated broader ethical impact.

**Jost Tobias Springenberg** guided Gato's deployment to the physical robot, provided strong existing baselines for block stacking, and advised on model development and experimental design.

**Tom Eccles** developed the Gato dialogue and image captioning demonstrations, allowing users to easily probe the vision and language capacities of agents in development.

**Jake Bruce** contributed to agent design as well as control datasets and environments with randomized physics and morphology variations.

**Ali Razavi** helped in exploring vision architectures.

**Ashley Edwards** contributed to the first prototype of Gato that worked on Atari, in addition to exploring alternative network architectures and training objectives.

**Nicolas Heess** advised on agent design, experiment design and task selection, especially for continuous control applications.

**Yutian Chen** advised on model design and experiments, and provided feedback in regular meetings.

**Raia Hadsell** advised on the design and planning of robotics efforts.

**Oriol Vinyals** advised on all aspects of the project, especially model architecture, training strategies and benchmark design.

**Mahyar Bordbar** was the primary project manager; eliciting key goals, tracking progress, facilitating presentations and feedback, and coordinating resource planning.

**Nando de Freitas** oversaw the project from its inception.

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

# Supplementary Material

## A   Model card

We present a model card for Gato in Table 4.

Table 4: **Gato Model Card.** We follow the framework proposed in (Mitchell et al., 2019).

| Model details | |
|---|---|
| Organization | DeepMind |
| Model Date | May 2022 |
| Model Type | Transformer with ResNet patch embedding for multi-task, multi-modal behavior cloning. |
| Model Version | Initial release. |
| Feedback on the Model | reedscot@google.com |
| **Intended Uses** | |
| Primary Intended Uses | Learn to accomplish a wide variety of tasks from expert demonstrations, such as playing video games, controlling simulated embodiments, and real world block stacking. |
| Primary Intended Users | DeepMind Researchers. |
| Out-of-Scope Uses | Not intended for commercial or production use. Military uses are strictly prohibited. |
| **Factors** | |
| Relevant Factors | Salient factors that may alter model performance are: agent embodiment in control data, training data token amount and diversity, performance of expert in training data and prompts (filtered by success rate), and any factors inherited by vision & language datasets described in Section 3.2. See Section 5.2, in particular Figure 9, for a detailed discussion of factors relating to training data diversity. |
| Evaluation Factors | Reported factors are: number of input tokens, proportion of data from different domains, agent performance. Many relevant factors are left for future work as use cases develop. |
| **Metrics** | |
| Model Performance Measures | We chose to report episode return for our control tasks. We decided not to report validation loss over held-out data because we found that it did not correlate well with episode return on the held-out tasks. |
| Decision thresholds | N/A |

| | |
|---|---|
| Approaches to Uncertainty and Variability | The reported values do not take into consideration model uncertainty as they are evaluations of a single model. It is prohibitive for us to collect the full suite of results with multiple models, however we have not observed statistically significant variations between different models evaluated on subsets of our benchmarks. We account for environment noise in the control tasks we use for evaluation by averaging returns across multiple episodes. To reduce variance introduced when selecting datasets of the limited demonstrations used during fine-tuning we generate 3 independent sets of datasets. The model is fine-tuned separately on each set of datasets and we take the mean performance across all of them. |
| **Evaluation Data** | |
| Datasets | Gato is evaluated on in and out of distribution simulated control tasks, see Section 4.1 and Section 5.2 for further details about these tasks. We also evaluated on the Skill Generalization challenge from the RGB Stacking robotics benchmark, see Section 4.2 and Section 5.3 for details. |
| Motivation | We evaluated on the in-distribution simulated control and robotics tasks to understand on how well Gato handles multi-modal and multi-task learning. We evaluated on out of distribution simulated control and robotics tasks to understand how well Gato can adapt to entirely new tasks. |
| Preprocessing | Observations from evaluation tasks are tokenized into a stream of discrete embeddings before being input to Gato. Section 2.1 and Section 2.2 go into details of how different modalities are tokenized and combined. |
| **Training Data** | |
| Datasets | We use a diverse and large number of datasets for training Gato. These include data from agent experience on both simulated and real world environments, along with a variety of natural language and image datasets. See Table 1 for details on our training datasets. |
| Motivation | To create a multi-modal, multi-task, multi-embodiment generalist policy we collected as much, diverse, data as possible. Joint training on all the datasets has produced a single network, Gato, which is capable of playing Atari, captioning images, chat, stacking blocks with a real robot arm, and more. See Section 3 for a more detailed discussion of our training datasets. |
| Preprocessing | The multi-modal training data is tokenized into a stream of discrete embeddings. Section 2.1 and Section 2.2 go into details of how different modalities are tokenized and combined. |
| **Quantitative Analyses** | |
| Unitary Results | We present several evaluations of Gato against different benchmarks. See Figure 5 for an analysis of Gato's performance on in distribution control tasks. Sections 5.2, 5.3, and 5.4 analyze performance on out of distribution control tasks. Finally, see Section 5.1 for a discussion on how model scale affects in-distribution performance. |
| **Ethical Considerations** | |
| Data | The vision and language datasets used include racist, sexist, and otherwise harmful context. |

| | |
|---|---|
| Risks and Harms | In addition to the potential harms of toxic image and language training data, Gato's real world embodiment introduces physical safety harms due to misuse or malfunctioning. |
| Mitigations | No mitigation of bias introduced by vision and language data beyond the filtering of sexually explicit content, as in Alayrac et al. (2022). Physical risk is mitigated through safety measures implemented by robotics environment designers. |
| **Caveats and Recommendation** | |
| Future work | The interaction of diverse training data domains and the different affordances faced in evaluation is poorly understood, and potential ethical and safety risks arise as the generalist's capabilities grow. |

## B    Agent Data Tokenization Details

In this section we provide additional details on our tokenization schemes. Our agent data is sequenced as follows:

- **Episodes** are presented to the agent in order of time (timesteps).

- **Timesteps** in turn are presented in the following order:
    - **Observations** ($[y_{1:k}, x_{1:m}, z_{1:n}]$) are ordered lexicographically by key, each item is sequenced as follows:
        * Text tokens ($y_{1:k}$) are in the same order as the raw input text.
        * Image patch tokens ($x_{1:m}$) are in raster order.
        * Tensors ($z_{1:n}$) (such as discrete and continuous observations) are in row-major order.
    - **Separator** ($'|'$); a designated separator token is provided after observations.
    - **Actions** ($a_{1:A}$) are tokenized as discrete or continuous values and in row-major order.

A full sequence of tokens is thus given as the concatenation of data from T timesteps:

$$s_{1:L} = [[y_{1:k}^1, x_{1:m}^1, z_{1:n}^1, '|', a_{1:A}^1], \ldots, [y_{1:k}^T, x_{1:m}^T, z_{1:n}^T, '|', a_{1:A}^T]],$$

where $L = T(k + m + n + 1 + A)$ is the total number of tokens.

Each floating point element of tensors in the observation sequence is mu-law companded as in WaveNet (Oord et al., 2016):

$$F(x) = \text{sgn}(x)\frac{\log(|x|\mu + 1.0)}{\log(M\mu + 1.0)} \tag{3}$$

with parameters $\mu = 100$ and $M = 256$. (If the floating-point tensor is in the action set, we do not need to compand the elements in the sequence because actions are only defined in the range $[-1, 1]$ for all our environments.) All the elements are subsequently clipped so that they fall in the set $[-1, 1]$. Finally, they are discretized using bins of uniform width on the domain $[-1, 1]$. We use 1024 bins and shift the resulting integers so they are not overlapping with the ones used for text tokens. The tokenized result is therefore a sequence of integers within the range of $[32000, 33024)$.

See Figure 14 and Figure 15 for visualizations of tokenizing and sequencing values (both discrete and continuous) and images. See Section C for details about local position encodings referenced in the figures.

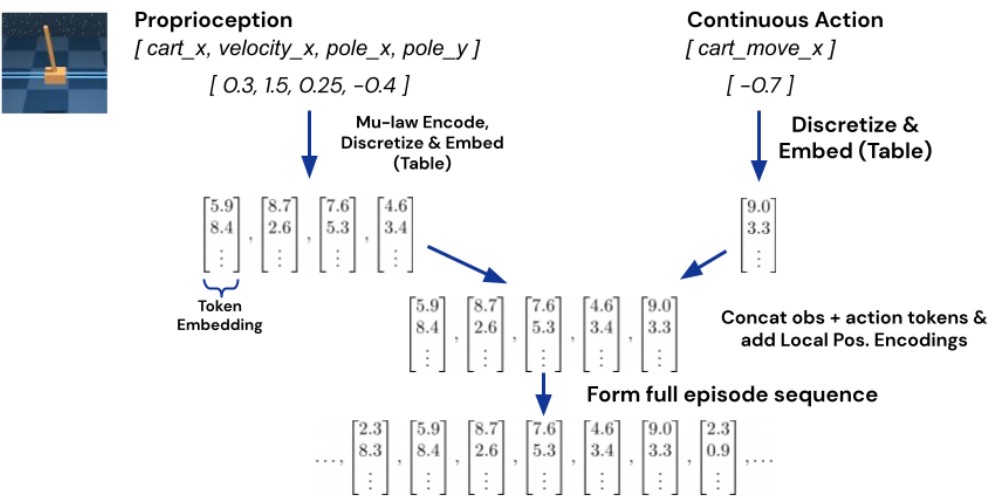

Figure 14: **A visualization of tokenizing and sequencing continuous values, e.g. proprioception.**

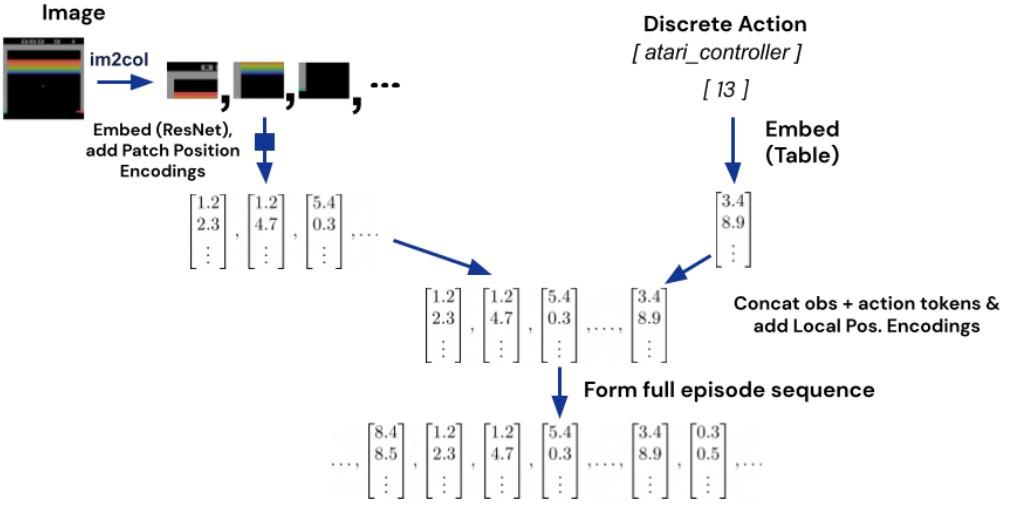

Figure 15: **A visualization of tokenizing and sequencing images and discrete values.**

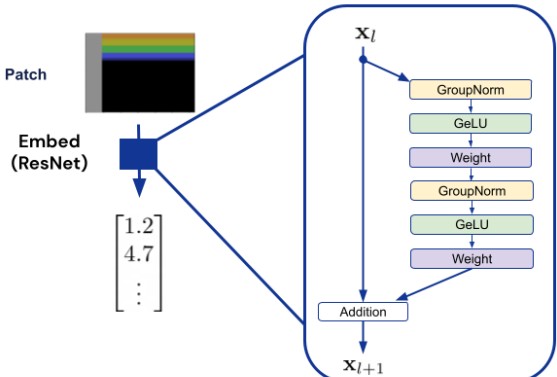

Figure 16: **Architecture of the ResNet block used to convert tokenized image patches into token embeddings.** This block uses the v2 ResNet architecture (He et al., 2016b), GroupNorm (Wu & He, 2018) (instead of LayerNorm (Ba et al., 2016)) normalization, and GELU (Hendrycks & Gimpel, 2016) (instead of RELU) activation functions.

# C   Model Architecture

## C.1   Transformer Hyperparameters

Table 5: **Gato transformer hyperparameters.**

| Hyperparameter | Gato 1.18B | 364M | 79M |
|---|---|---|---|
| TRANSFORMER BLOCKS | 24 | 12 | 8 |
| ATTENTION HEADS | 16 | 12 | 24 |
| LAYER WIDTH | 2048 | 1536 | 768 |
| FEEDFORWARD HIDDEN SIZE | 8192 | 6144 | 3072 |
| KEY/VALUE SIZE | 128 | 128 | 32 |
| SHARED EMBEDDING | TRUE | | |
| LAYER NORMALIZATION | PRE-NORM | | |
| ACTIVATION FUNCTION | GEGLU  (SHAZEER, 2020) | | |

The transformer hyperparameters of Gato are presented in Table 5. We also list the hyperparameters of smaller architecture variants used in Section 5.

## C.2   Embedding Function

The ResNet block uses the v2 architecture (He et al., 2016b), contains GroupNorm (Wu & He, 2018) with 32 groups instead of LayerNorm (Ba et al., 2016), and GELU (Hendrycks & Gimpel, 2016) activation functions instead of RELU. The block is diagrammed in Figure 16.

## C.3   Position Encodings

After tokens are mapped into token embeddings, two position encodings are added to the token embeddings (when applicable) to provide temporal and spatial information to the model. These are described below.

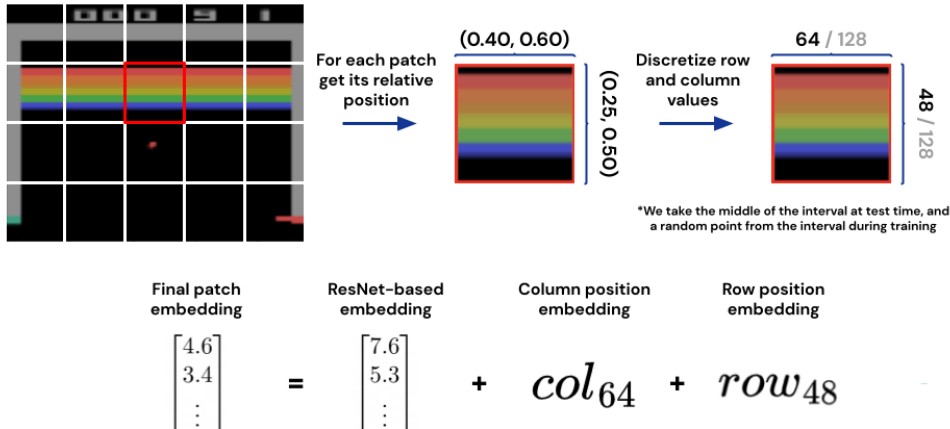

Figure 17: **Patch position encodings.** Calculating patch position encodings (red) within the global image (far left). The relative row and column positions (i.e. positions normalized between $[0, 1]$) are first discretized using uniform binning and used to index a learnable row and column position encoding. These two encodings are then added to the token embedding corresponding to the patch.

### Patch Position Encodings

These position encodings convey information about a patch's global position within the image from which the patch was extracted. First, the relative row and column intervals of the patch are calculated by normalizing the patch's pixel intervals by the image resolution. The row and column normalized intervals are then quantized into a vocabulary size (we use 128) and are used to index a row and column table of learnable position encodings. The method in which the quantized row and column intervals are converted into indices depends on whether we are training or evaluating the model: during training a random index is uniformly sampled from the quantized interval, while during evaluation we deterministically take the (rounded) mean of the interval. Once row and column position encoding are retrieved from the embedding table, they are added onto the token embedding produced by the resnet embedding function, as described previously.

To more concretely demonstrate this process, we provide an example in Figure 17. We will follow the process with the patch highlighted in red on the left of the subfigure. The image is of resolution $80 \times 64$ and each patch is $16 \times 16$, meaning there are $5 \times 4 = 20$ patches total. The highlighted patch starts at pixel row interval $[16, 32]$ and pixel column interval $[32, 64]$. Normalized, the row interval is therefore $[0.25, 0.5]$ and the column interval is $[0.4, 0.6]$. We then separately quantize the intervals into 128 uniformly spaced bins, with the resulting quantized row interval being $[32, 64]$ and the quantized column interval being $[51, 77]$. During training, we uniformly sample integers between the quantized row intervals, whereas during testing we would use the means, which are index 48 for row position and index 64 for column position. The row and column positions are finally used to index separate row and column position encoding tables to produce learnable embeddings which are added onto the corresponding patch token embedding.

### Local Observation Position Encodings

The local observation position encoding adds positional information about where observation tokens are positioned within the local time-step they were an element of. First, we reiterate that, during tokenization, for each time-step all elements of the observation set are tokenized into sequences and concatenated into an observation sequence. Each token in this observation sequence is given an index which corresponds to the sequence order, i.e. the first token is 0 and the last is the length of the observation sequence minus one. After embedding, for any tokens that were a part of an observation set, the corresponding observation token

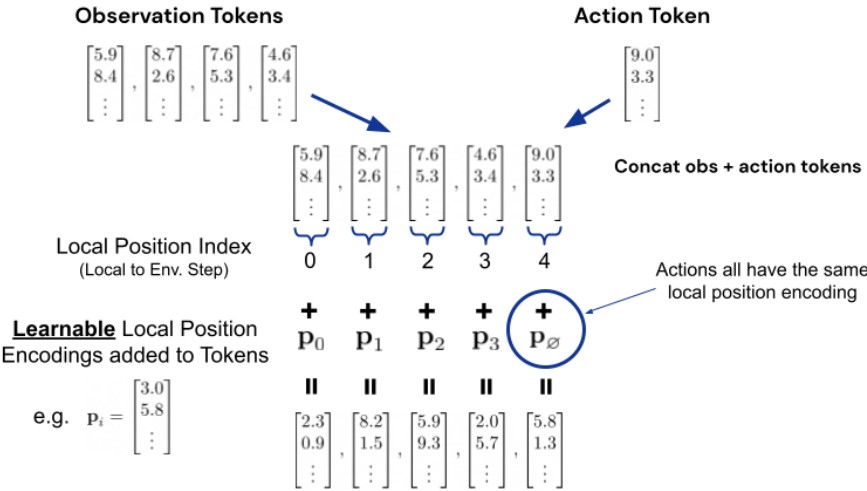

Figure 18: **Local position encodings.** An example demonstrating how local position encodings are defined within each time-step's observation and action token subsequences. Note that no position encodings are added to action tokens.

index is used to index an embedding table of learnable position encodings, with one embedding for every possible observation token index (in practice we simply set the table size to a large value like 512). The position encoding is then added onto the observation token embedding to produce the final token embedding. Note that all action tokens are given the same position encoding regardless of their position in the time-step sequence. We illustrate an example of this process in Figure 18.

## D Pretraining Setup

**Optimizer:** For all models we use the AdamW (Loshchilov & Hutter, 2017) optimizer with a linear warm-up and cosine schedule decay. The linear warmup lasts for $15,000$ steps, starting from a learning rate of `1e-7` and ending at a different maximum learning rate depending on the model (see Table 6). This learning rate is then cosine decayed by a factor 10x over 1,000,000 steps. The AdamW optimizer has parameters $\beta_1 = 0.9$, $\beta_2 = 0.95$ and $\epsilon = $ `1e-8`. We use a batch size of 512 and a sequence length of 1024 tokens for all models.

**Regularization:** We train with an AdamW weight decay parameter of 0.1. Additionally, we use stochastic depth (Huang et al., 2016) during pretraining, where each of the transformer sub-layers (i.e. each Multi-Head Attention and Dense Feedforward layer) is skipped with a probability of 0.1.

Table 6: **Learning rate schedule hyperparameters for the different model scales.**

| Hyperparameter | Gato 1.18B | 364M | 79M |
|---|---|---|---|
| MAXIMUM LEARNING RATE | 1E-4 | 2E-4 | 1E-4 |
| MINIMUM LEARNING RATE | 1E-5 | 2E-5 | 1E-5 |

## E    Fine-tuning Setup

**Optimizer:**   For all models we use the Adam (Kingma & Ba, 2014) optimizer with a constant learning rate of `1e-5`. The Adam optimizer has parameters $\beta_1 = 0.9$, $\beta_2 = 0.95$ and $\epsilon = $ `1e-8`. We use a batch size of 64 and a sequence length of 1024 tokens for all models. We train for 10,000 gradient steps.

**Regularization:** We use dropout (Srivastava et al., 2014) with a rate of 0.1.

**Evaluation:** We evaluate agent every 100 learning steps. Each evaluation reports the average of 10 runs of a given checkpoint. The moving average of 5 such scores is computed (to gather 50 runs together). The final fine-tuning performance is defined as the maximum of these smoothed scores.

**Datasets:** We generated data for the fine-tuning tasks the same way we did for the other tasks (see Section 3.1 for details). Instead of using all the data for a fine-tuning task, we discarded all but 2000 best episodes (leading to the highest returns). The fine-tuning datasets were created in the following way. We randomly took 1000 episodes (out of 2000 preselected episodes), then a subset of 100 episodes from the selected episodes, then 10, 5, 3, and finally a single episode. We repeated this procedure 3 times to obtain 3 series of cascading subsets for each task. Each subset is used to conduct one fine-tuning experiment, and each is reported on our plots in Section 5.2 as a separate point.

**Task settings:** We have not altered any of the tasks and used their canonical versions. As 3 out of 4 tasks are open sourced, they do not need further explanation. For the fourth task, DMLab `order_of_apples_forage_simple`, the goal is to collect apples in the right order, green ones first followed by the gold one.

## F    Data Collection Details

### F.1    Atari

We collect two separate sets of Atari environments. The first (that we refer to as ALE Atari) consists of 51 canonical games from the Arcade Learning Environment (Bellemare et al., 2013). The second (that we refer to as ALE Atari Extended) is a set of alternative games[3] with their game mode and difficulty randomly set at the beginning of each episode.

For each environment in these sets we collect data by training a Muesli (Hessel et al., 2021) agent for 200M total environment steps. We record approximately 20,000 random episodes generated by the agent during training.

### F.2    Sokoban

Sokoban is a planning problem (Racanière et al., 2017), in which the agent has to push boxes to target locations. Some of the moves are irreversible and consequently mistakes can render the puzzle unsolvable. Planning ahead of time is therefore necessary to succeed at this puzzle. We use a Muesli (Hessel et al., 2021) agent to collect training data.

### F.3    BabyAI

BabyAI is a gridworld environment whose levels consist of instruction-following tasks that are described by a synthetic language. We generate data for these levels with the built-in BabyAI bot. The bot has access to extra information which is used to execute optimal solutions, see Section C in the appendix of (Chevalier-Boisvert et al., 2018) for more details about the bot. We collect 100,000 episodes for each level.

---

[3]Basic Math, Breakout, Crossbow, Darkchambers, Entombed, ET, Flag Capture, Human Cannonball, Klax, Laser Gates, Ms. Pac-Man, Solaris, Space War.

### F.4 DeepMind Control Suite

The DeepMind Control Suite (Tunyasuvunakool et al., 2020; Tassa et al., 2018) is a set of physics-based simulation environments. For each task in the control suite we collect two disjoint sets of data, one using only state features and another using only pixels. We use a D4PG (Barth-Maron et al., 2018) agent to collect data from tasks with state features, and an MPO (Abdolmaleki et al., 2018) based agent to collect data using pixels.

We also collect data for randomized versions of the control suite tasks with a D4PG agent. These versions randomize the actuator gear, joint range, stiffness, and damping, and geom size and density. There are two difficulty settings for the randomized versions. The small setting scales values by a random number sampled from the union of intervals $[0.9, 0.95] \cup [1.05, 1.1]$. The large setting scales values by a random number sampled from the union of intervals $[0.6, 0.8] \cup [1.2, 1.4]$.

### F.5 DeepMind Lab

DeepMind Lab (Beattie et al., 2016) is a first-person 3D environment designed to teach agents 3D vision from raw pixel inputs with an egocentric viewpoint, navigation, and planning.

We trained an IMPALA (Espeholt et al., 2018) agent jointly on a set of 18 parent DM Lab levels that generate maps procedurally for each new episode. Data was collected by executing the agent on these 18 levels, as well as an additional set of 237 levels handcrafted to test a diverse set of skills.

The 18 parent levels are characterized by high diversity of generated maps. The difference between the levels is rooted in hyper-parameters used in a generation process. These hyper-parameters control high-level characteristics such as types of structures spawned, difficulty of language instructions, or presence of specific tools. The parent levels were developed to improve performance of RL agents trained online on them.

In contrast to the parent levels, each of the additional handcrafted 237 levels uses almost the same map, and the main differences between instances of the same level map are aesthetics such as colors of walls or lighting conditions. The maps are *not* procedurally generated and were designed to test a diverse set of skills such as walking up stairs or using specific tools. They are similar to levels presented in Figure 3, Figure 7 and Figure 8 in aforementioned paper by Beattie et al. (2016).

Additional information on the 18 parent levels (and their relation to the other levels) is presnted in details in the NeurIPS Workshop talk *A Methodology for RL Environment Research* by Daniel Tanis[4].

In total, we collected data for 255 levels from the DeepMind Lab (18 parent levels and 237 handcrafted levels), 254 of which were used while training Gato. The remaining level was used for out of distribution evaluation.

### F.6 Procgen Benchmark

Procgen (Cobbe et al., 2020) is a suite of 16 procedurally generated Atari-like environments, which was proposed to benchmark sample efficiency and generalization in reinforcement learning. Data collection was done while training a R2D2 (Kapturowski et al., 2018) agent on each of the environments. We used the hard difficulty setting for all environments except for maze and heist, which we set to easy.

### F.7 Modular RL

Modular RL (Huang et al., 2020) is a collection of MuJoCo (Todorov et al., 2012) based continuous control environments, composed of three sets of variants of the OpenAI Gym (Brockman et al., 2016) Walker2d-v2, Humanoid-v2, and Hopper-v2. Each variant is a morphological modification of the original body: the set of

---

[4]Available at `https://neurips.cc/virtual/2021/workshop/21865#wse-detail-22801`.

morphologies is generated by enumerating all possible subsets of limbs, and keeping only those sets that a) contain the torso, and b) still form a connected graph. This results in a set of variants with different input and output sizes, as well as different dynamics than the original morphologies. We collected data by training a single morphology-specific D4PG agent on each variant for a total of 140M actor steps, this was done for 30 random seeds per variant.

### F.8 DeepMind Manipulation Playground

The DeepMind Manipulation Playground (Zolna et al., 2021) is a suite of MuJoCo based simulated robot tasks. We collect data for 4 of the Jaco tasks (box, stack banana, insertion, and slide) using a Critic-Regularized Regression (CRR) agent (Wang et al., 2020) trained from images on human demonstrations. The collected data includes the MuJoCo physics state, which is we use for training and evaluating Gato.

### F.9 Meta-World

Meta-World (Yu et al., 2020) is a suite of environments[5] for benchmarking meta-reinforcement learning and multi-task learning. We collect data from all train and test tasks in the MT50 mode by training a MPO agent (Abdolmaleki et al., 2018) with unlimited environment seeds and with access to state of the MuJoCo physics engine. The collected data also contains the MuJoCo physics engine state.

## G  Real robotics evaluation details

In the real world, control is asynchronous; physics does not wait for computations to finish. Thus, inference latency is a concern for evaluating a large model for real world tasks. In robotics, a fast control rate is thought to be critical for reacting to dynamic phenomena. The robot setup for RGB stacking has a 20Hz control rate (0.05 second timestep) by design. In order to reach an acceptable margin of latency, we modified inference at evaluation time by shortening the context length to 1. We also implemented a parallel sampling scheme where all the action tokens are zeroed out in the input sequences during training so we can sample all tokens corresponding to a robot action in a single model inference step instead of autoregressively as it's done in other domains. We found that the `1.18B` parameter model was able to run on the hardware accelerators in our robots (NVidia GeForce RTX 3090s), but still overran the 20Hz control rate by a small amount ($\sim$0.01 seconds).

We use the sparse reward function described in Lee et al. (2021) for data filtering. We only select trajectories with *final* task success; that is, a sparse reward of 1 on the final timestep.

## H  Skill Mastery architecture

The numbers reported for the Skill Mastery benchmark were collected by executing a model zero-shot that used an earlier version of the Gato architecture. Instead of the ResNet patch embedding, a similar architecture using a local transformer was used to embed image patch tokens. The local position embeddings and patch position embeddings were not used. These changes were implemented and found to improve Gato's performance after the pretraining data was changed (as we decided to focus on Skill Generalization instead of Skill Mastery challenge), which is why they are presented as the final architecture of our full model.

---

[5]We used a version from July 23rd 2021, specifically the following version: `https://github.com/rlworkgroup/metaworld/commit/a0009ed9a208ff9864a5c1368c04c273bb20dd06`.

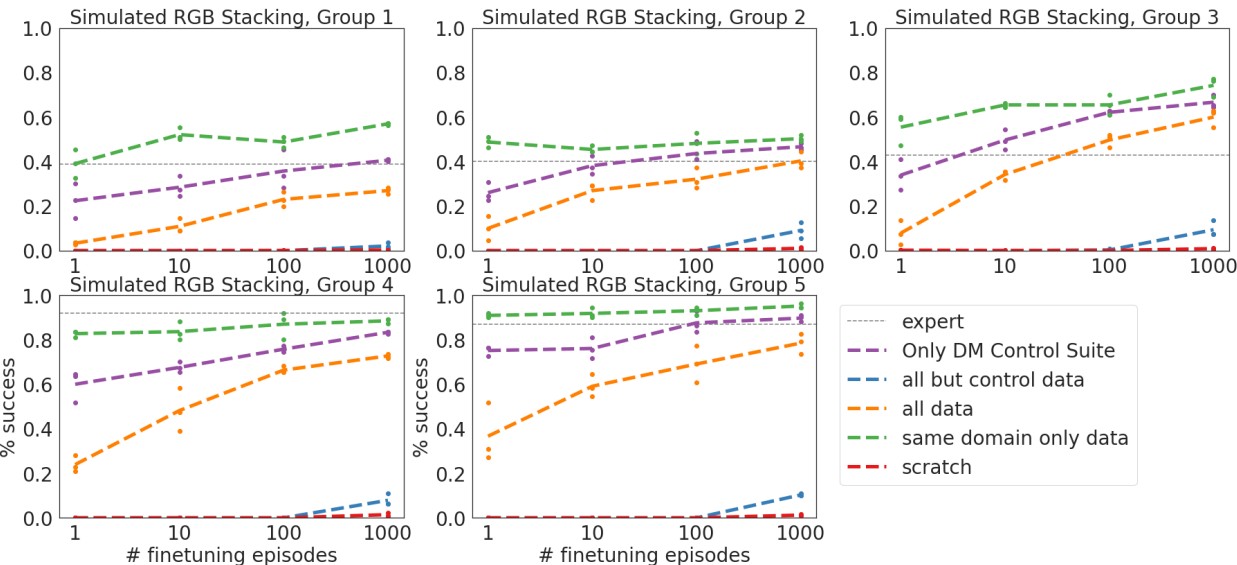

Figure 19: **Few-shot performance of Gato for Skill Generalization in simulation.** Each test set object is plotted separately. We ablate over different pretraining datasets.

## I  Additional robotics ablations

We conducted a series of ablations in simulation to better understand the effect of diverse pretraining data in the robotics domain (see Figure 19). We included the same baselines as in Section 5.2, selecting the `364M` parameter size variant, as well as an additional baseline trained with control suite data only. The DM Control-only agent is superior to the base Gato at zero-shot transfer and with a lot of fine-tuning data, suggesting that Gato may not be using the representations learned from the text-based datasets when adapting to robotics tasks. The same domain only agent performs the best overall, matching the CRR baseline at 1 fine-tuning episode and outperforming it with more data, suggesting that Gato at current scale can trade its generalization capacity for data-efficient and effective few-shot adaptation.

## J  Attention visualization

To render the transformer attention weights, we retrieved the cross-attention logits, a tensor with dimension $(H, T, T)$ where $H$ is the number of heads and $T$ is the number of tokens in a sequence. The $(h, i, j)$th entry of this matrix can be interpreted as the amount that head $h$ attends to token $j$ from token $i$. Due to Gato's image tokenization scheme, there are multiple tokens per timestep. Therefore to render the attention for a particular timestep, we took the sub-matrix that corresponds to that timestep. We then applied a softmax over the rows of this matrix to normalize the relevant values. Because we are only interested in attention to the previous tokens, we excluded the diagonal by setting it to negative infinity before softmax.

To measure the importance of each patch, we averaged the attention weights over the corresponding column. Because Gato uses a causal transformer, the attention matrix is lower triangular, so the mean was only considered over the sub-column below the diagonal of the matrix. This corresponds to the average attention paid to particular patch over a whole timestep.

Using this method, we found the attention maps at the first layer the transformer to be most interpretable, agreeing with the findings of Abnar & Zuidema (2020). Certain heads clearly track task-specific entities and regions of the image. Figure 20 shows the attention maps for manually selected heads at the first layer for several tasks.

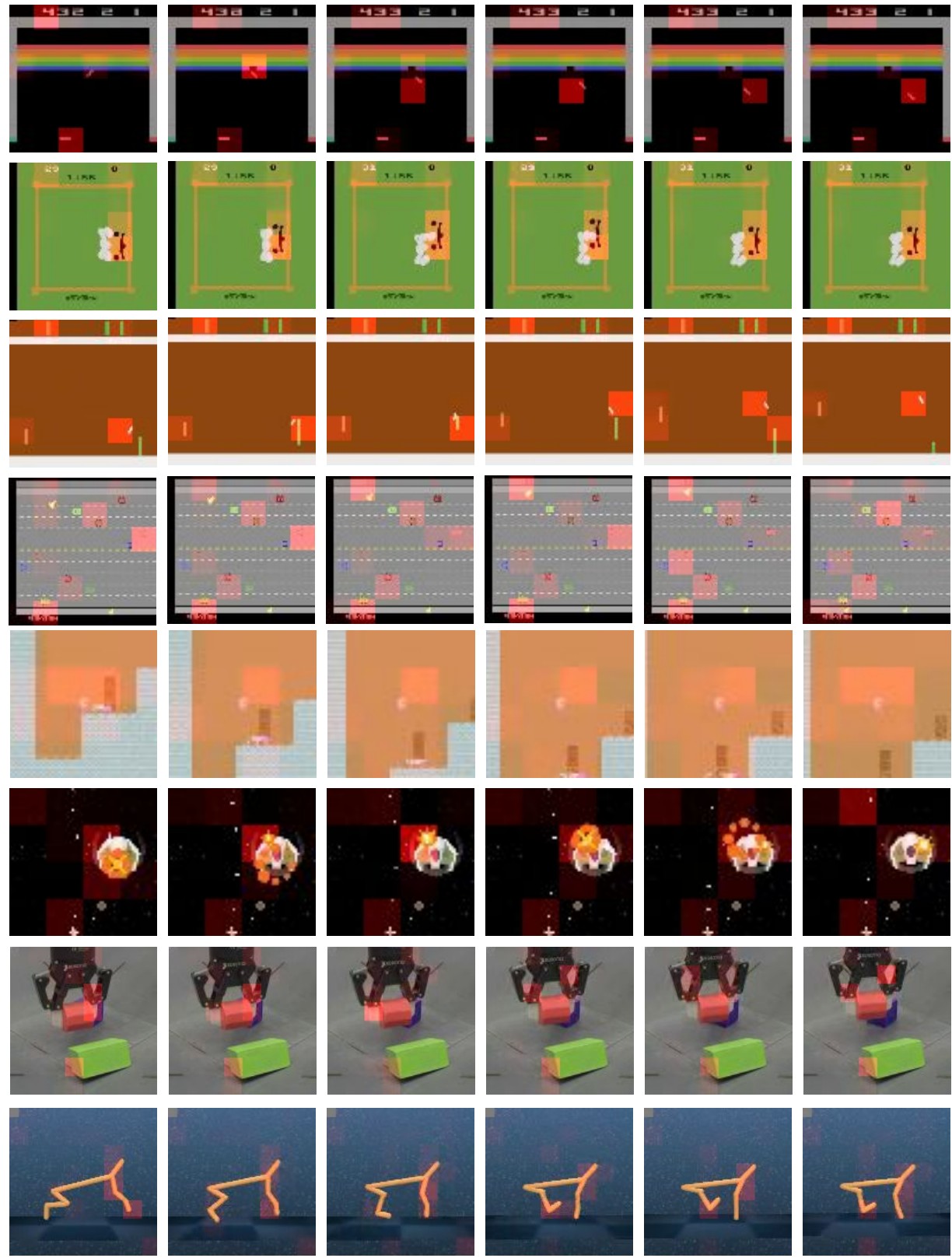

Figure 20: **Attention maps.** Time-lapse attention maps from selected heads at the first layer for Atari Breakout, Boxing, Pong, Freeway, Procgen CoinRun, Bossfight, RGB Stacking, and DM Control Suite Cheetah.

## K    Detailed results for specialist Meta-World agent

The specialist Meta-World agent described in Section 5.5 achieves 96.6% success rate averaged over all 50 Meta-World tasks. The detailed success rates are presented in Table 7. We evaluated agent 500 times for each task.

Table 7: **Success rates of specialist Meta-World agent.** Averaged over 500 evaluations.

| Task name | Success rate |
|---|---|
| ASSEMBLY-V2 | 0.980 |
| BASKETBALL-V2 | 0.964 |
| BIN-PICKING-V2 | 0.954 |
| BOX-CLOSE-V2 | 0.958 |
| BUTTON-PRESS-TOPDOWN-V2 | 0.996 |
| BUTTON-PRESS-TOPDOWN-WALL-V2 | 0.998 |
| BUTTON-PRESS-V2 | 0.996 |
| BUTTON-PRESS-WALL-V2 | 1.000 |
| COFFEE-BUTTON-V2 | 1.000 |
| COFFEE-PULL-V2 | 0.980 |
| COFFEE-PUSH-V2 | 0.974 |
| DIAL-TURN-V2 | 0.916 |
| DISASSEMBLE-V2 | 0.924 |
| DOOR-CLOSE-V2 | 0.994 |
| DOOR-LOCK-V2 | 0.986 |
| DOOR-OPEN-V2 | 1.000 |
| DOOR-UNLOCK-V2 | 0.994 |
| DRAWER-CLOSE-V2 | 1.000 |
| DRAWER-OPEN-V2 | 0.992 |
| FAUCET-CLOSE-V2 | 0.982 |
| FAUCET-OPEN-V2 | 0.996 |
| HAMMER-V2 | 0.998 |
| HAND-INSERT-V2 | 0.960 |
| HANDLE-PRESS-SIDE-V2 | 0.972 |
| HANDLE-PRESS-V2 | 0.946 |
| HANDLE-PULL-SIDE-V2 | 0.992 |
| HANDLE-PULL-V2 | 0.992 |
| LEVER-PULL-V2 | 0.980 |
| PEG-INSERT-SIDE-V2 | 0.992 |
| PEG-UNPLUG-SIDE-V2 | 0.994 |
| PICK-OUT-OF-HOLE-V2 | 0.966 |
| PICK-PLACE-V2 | 0.990 |
| PICK-PLACE-WALL-V2 | 0.986 |
| PLATE-SLIDE-BACK-SIDE-V2 | 1.000 |
| PLATE-SLIDE-BACK-V2 | 0.994 |
| PLATE-SLIDE-SIDE-V2 | 1.000 |
| PLATE-SLIDE-V2 | 0.984 |
| PUSH-BACK-V2 | 0.984 |
| PUSH-V2 | 0.944 |
| PUSH-WALL-V2 | 0.784 |
| REACH-V2 | 0.796 |
| REACH-WALL-V2 | 0.802 |
| SHELF-PLACE-V2 | 0.958 |
| SOCCER-V2 | 0.968 |
| STICK-PULL-V2 | 0.882 |
| STICK-PUSH-V2 | 0.966 |
| SWEEP-INTO-V2 | 0.962 |
| SWEEP-V2 | 0.948 |
| WINDOW-CLOSE-V2 | 1.000 |
| WINDOW-OPEN-V2 | 1.000 |
| **Average** | **0.966** |

## L    Per-domain results for Gato

We describe performance of Gato for simulated control tasks in Section 4.1. In Table 8, we present normalized per-domain results. We evaluated agent 50 times for each task.

Table 8: **Normalized Gato per-domain scores.** Averaged over 50 evaluations.

| Control environment | Normalized Score (in %) |
|---|---|
| DM Lab | 91.4 |
| ALE Atari | 30.9 |
| ALE Atari Extended | 57.8 |
| Sokoban | 68.0 |
| BabyAI | 93.2 |
| DM Control Suite | 63.6 |
| DM Control Suite Pixels | 26.3 |
| Meta-World | 87.0 |
| Procgen Benchmark | 60.8 |
| RGB Stacking simulator | 58.0 |
| Modular RL | 62.9 |
| DM Manipulation Playground | 83.8 |

