# OpenReview forum: "A Generalist Agent"
_TMLR — Accepted by TMLR_

### Review · Reviewer_qCeZ · 2022-09-28

**Summary Of Contributions:**

This paper proposes a generalist agent, which is trained on large numbers of datasets via supervised learning so that this agent can play Atari, caption images, chat,  control robots, and so on.

**Requested Changes:**

Please see the pros and cons part.

In addition,

How Gato benefits from prompt should be analyzed in detail.

How to analyze the task interference in Gato should be discussed.

I am wondering about the results of using Gato to train on RL tasks via offline RL or even online learning.

In Section 2.3, the mask on the training loss only considers the token as text or action, which is inconsistent with the last paragraph in Section 2.2. This should be clarified.

Some descriptions are not convincing to me, for example, in Section 5.5, the results on specialist single-domain are very well, however, these are obtained by imitating a pre-trained RL agent, so this should be clarified: 'there is no single multi-task agent that approaches 100% success rate', I am assuming here single multi-task agent is an RL agent, but single and multi-task are conflicting.

**Strengths And Weaknesses:**

Strengths:

1) The biggest highlight of Gato is that a big model (transformer) can complete a large number of multi-modal and different types of tasks. Gato exceeds ordinary multi-task learning in terms of quantity and settings, using one network model with the same weights, accomplishing such a large number of tasks.
2) Extensive experiments and analysis provide sufficient empirical evidence.

Weaknesses:

1) The essence is imitation learning, for natural language or image datasets, it may be easy to obtain. However, for RL tasks, the data is obtained from SOTA or near-SOTA pre-trained RL agents, which may be non-trivial and take a lot of time.

2) Although Gato is able to achieve multi-model multi-task training, it does not bring a significant improvement in a single task compared with big models in the field of NLP.

3) The generalization ability in a specific field works well, but it highly depends on the successful demonstration data as Prompt which needs to be further explored.

4) When a single network with the same weights is used to process a huge number of tasks, various conflicts/interference will inevitably arise among tasks. Knowledge of some tasks will not promote the learning of other tasks but will hinder learning, which is not considered in this paper. For example, the knowledge learned on many image datasets does not promote the learning on Atari or DMC.

At present, Gato uses a purely supervised learning method, which is somewhat similar to trajectory transformers and decision transformers(also mentioned in the related work). However, in many continuous control tasks, RL may be better.

Several experimental results in this paper try to prove that the larger the parameter, the better the performance. The current model parameters are only 1.2 billion. I am wondering if it is expanded to 120 billion and more tasks are added, will it be able to achieve even better results?

---

> ### Author Response · Authors · 2022-10-07
> **Response to the reviewer qCeZ [part 1]**
>
> Thank you for the encouraging review.
>
> Prompted by TMLR reviews, we added a new section to the revised version of our submission: Section 8 “Limitations and Future work”. We will refer to this section twice in our response here.
>
> In the following we hope to address all your concerns one by one. If there is anything missing, please let us know.
>
>   > The essence is imitation learning, for natural language or image datasets, it may be easy to obtain. However, for RL tasks, the data is obtained from SOTA or near-SOTA pre-trained RL agents, which may be non-trivial and take a lot of time.
>
> Gato is a data-driven approach, as it is derived from imitation learning. While natural language or image datasets are relatively easy to obtain from the web, a web-scale dataset for control tasks is not currently available. This may seem at first to be problematic, especially when scaling Gato to a higher number of parameters.
>
> That being said, there has already been extensive investigation into this issue. Offline RL aims at leveraging existing control datasets, and its increasing popularity has already resulted in the availability of more diverse and larger datasets. Richer environments and simulations are being built (e.g. Metaverse), and increasing numbers of users already interact with them among thousands of already deployed online games (e.g. there exists a large dataset of Starcraft 2 games). Real-life data has also been already stored for ML research purposes; for example, data for training self-driving cars is acquired from recording human driver data. Finally, while Gato uses data consisting of both observations and corresponding actions, the possibility of using large scale observation-only data to enhance agents has been already studied [Baker et al. 2022]. Thanks to online video sharing and streaming platforms such as Youtube and Twitch, observation-only datasets are not significantly more difficult to collect than natural language datasets, motivating a future research direction to extend Gato to learn from web data.
>
> While the previous paragraph focuses on alleviating drawbacks of data collection from RL agents, it is important to note that this approach presents a different set of tradeoffs compared to scraping web data and can be actually more practical in some situations. Once the simulation is set up and near SOTA agent trained, it can be used to generate massive amounts of high quality data. That is in contrast to the quality of web data which is notorious for its low quality.
>
> In short, we believe that acquiring suitable data is another research question on its own, and this is an active area of research with growing momentum and importance.
>
> We discussed this matter in the new section added to the paper (mentioned in the beginning of our response).
>
>   > Although Gato is able to achieve multi-model multi-task training, it does not bring a significant improvement in a single task compared with big models in the field of NLP.
>
> Gato focuses on multi-task training and it achieves near expert/SOTA scores for a few hundreds of tasks. The specialist Meta-World agent (see Section 5.5) approaches 100% success rate and, to the best of our knowledge, it is the best model that does that.

---

> ### Author Response · Authors · 2022-10-07
> **Response to the reviewer qCeZ [part 2]**
>
>
>   > The generalization ability in a specific field works well, but it highly depends on the successful demonstration data as Prompt which needs to be further explored.
>
>   > How Gato benefits from prompt should be analyzed in detail.
>
> Gato is prompted with an expert demonstration, which aids the agent to output actions corresponding to the given task. This is particularly useful since there is otherwise no task identifier available to the agent (that is in contrast to many multi-task RL settings). Gato infers the relevant task from the observations and actions in the prompt.
>
> However, the context length of our agent is limited to 1024 tokens which translates to the agent sometimes attending to only a few environment timesteps in total. This is especially the case for environments with image observations, where depending on the resolution each observation can result in more than one hundred tokens each. Hence for certain environments only a short chunk of a demonstration episode fits in the transformer memory.
>
> Due to this limited prompt context, preliminary experiments with different prompt structures resulted in very similar performance. Similarly, early evaluations of the model using prompt-based in-context learning on new environments did not show a significant performance improvement compared to prompt-less evaluation in the same setting.
>
> Context-length is therefore a current limitation of our architecture, mainly due to the quadratic scaling of self-attention. Many recently proposed architectures enable a longer context at greater efficiency and these innovations could potentially improve our agent performance. We hope to explore these architectures in future work.
>
> We discussed this matter in the new section added to the paper (mentioned in the beginning of our response).
>
>   > When a single network with the same weights is used to process a huge number of tasks, various conflicts/interference will inevitably arise among tasks. Knowledge of some tasks will not promote the learning of other tasks but will hinder learning, which is not considered in this paper. For example, the knowledge learned on many image datasets does not promote the learning on Atari or DMC.
>
>   > How to analyze the task interference in Gato should be discussed.
>
> While we believe that such problems will hopefully disappear at the larger scale, we agree with the reviewer that it is currently the case. As you . Our experiment with specialist agents (Section 5.5) touches on these issues, as the model trained only on ALE Atari tasks outperforms Gato that is trained on the full suite of our data. Would you like us to elaborate a bit more about the problem in this section?
>
>   > At present, Gato uses a purely supervised learning method, which is somewhat similar to trajectory transformers and decision transformers(also mentioned in the related work). However, in many continuous control tasks, RL may be better.
>
>   > I am wondering about the results of using Gato to train on RL tasks via offline RL or even online learning.
>
> We decided to focus on pure supervised learning, as it already posed us quite a challenge. However, we believe that adding online RL finetuning would unlock new possibilities of adapting to completely new tasks. That would also address your first point about lack of data for many RL tasks. Incorporating offline RL techniques can also be beneficial.
>
>   > Several experimental results in this paper try to prove that the larger the parameter, the better the performance. The current model parameters are only 1.2 billion. I am wondering if it is expanded to 120 billion and more tasks are added, will it be able to achieve even better results?
>
> We purposely restricted Gato’s size such that it can be run in real-time on the real robot. We hope that new architectures and hardware will allow us to scale to larger models. We would also like to see how the performance scales with more parameters and tasks.
>
>   > In Section 2.3, the mask on the training loss only considers the token as text or action, which is inconsistent with the last paragraph in Section 2.2. This should be clarified.
>
> Good catch. Corrected.
>
>   > Some descriptions are not convincing to me, for example, in Section 5.5, the results on specialist single-domain are very well, however, these are obtained by imitating a pre-trained RL agent, so this should be clarified: 'there is no single multi-task agent that approaches 100% success rate', I am assuming here single multi-task agent is an RL agent, but single and multi-task are conflicting.
>
> Thank you. We meant that there is no previously published single agent that approaches an average 100% success rate simultaneously for all tasks, and our agent is the first one to accomplish it. Corrected.

---

### Review · Reviewer_pzPk · 2022-09-29

**Summary Of Contributions:**

The idea behind the paper is quite simple. Given sequence modeling has been extremely successful in language modeling especially as we scale up the data and compute, Gato answers the question: what if we started scaling up behavior cloning which can also be framed as a sequence modeling problem. Other than the standard issues with context window length of sequence models the biggest bottleneck is the problem of encoding these various modalities into a single vector space of "tokens" that can be ingested by a large sequence model, often transformers. To this end, the paper proposes various "Tokenization" approaches to capture the fair amount of multi-modal data available from current crop of reinforcement learning benchmarks, together with relatively standard vision and language datasets.

Overall I see the major point being the importance of scale and large amounts of data for better agent behavior.

**Broader Impact Concerns:**

Broader impact concerns were sufficiently discussed.

**Requested Changes:**

I'm sure there are endless experiments one could do to analyze the nature of representations in the attention layers and various choices one could make for various parameters. Unless there are plan on sharing the code base, it would be great to add more concrete details on the finetuning data and settings for each task. Similarly it would have been great if the paper was more explicit about its limitations even if their hypothesis is "scale is all you need".

Minor questions:
- Appendix C, Table 5: Activation function is supposed to be GeLU?
- Why use AdamW for training but Adam for finetuning? No weight decay?
- What version of Meta-world? There are quite a few issues in various versions of the benchmark.
- How was the sample weight decided for various datasets? What is the individual dataset sizes in terms of "bytes".
- How did you stop training? Were any metrics tracked that can show convergence on any tasks?

**Strengths And Weaknesses:**

**Strengths**
The paper makes it fairly obvious that multi-task learning is actually a good way to get vastly more data. Issues of catastrophic forgetting and negative transfer largely go away at enough scale too. The real trick is in tokenization and details in Appendix B and Appendix C are super useful and would go a long way in helping other multi-modal researchers. While the ideas presented are in a way quite straightforward, paper is very clearly written and kudos for the sheer variety of tasks considered and doing justice to the title in a way.  I'm sure writing the dataprocessing and dataloaders were the hardest part of the endeavour and it's a shame that effort can't be shared in a paper easily. The paper also implies that future research requires focusing a lot more on obtaining the right data at right scale compared to model tuning.

**Weaknesses**
While the paper has many weaknesses which I try to list here, the biggest question in a sense can we even achieve the scale required for effective agents with this approach. While the paper says yes, it doesn't really contribute as much in how except the ability to deal with multi-modal multi-task data under a single framework. However, the data included thousands (if not millions) of episodes on various tasks which is largely only possible with simulations and simulated agents. For the current tasks, the authors likely had access to previously trained checkpoints from existing papers on thee existing benchmarks. However for novel tasks one would require building new simulations as well as figuring out a useful control agent that still gives enough diverse data. (Robot task it seems had 350k trajectories and still achieved only 50% performance) This is excarberated by the effect that it's unclear whether the "generalist agent" really learned to generalize. Compared to [1] it is a bit surprising that seeing _so_ much data didn't help more in having better representations for finetuning.  The evaluated tasks are relatively few and simple and it already had negative transfer on ALE. Moreover there wasn't much evaluation on frozen weight zero-shot performance on new tasks as has been demonstrated with various large language models. Could have been useful to analyze how exactly the information from different tasks are being stored in the distributed representations, is it more clustered or more [superimposed](https://transformer-circuits.pub/2022/toy_model/index.html) and whether Mixture-of-Experts style scaling might be even more effective. Similarly, would have been interesting to analyze the role of context window for various tasks and whether improving that can alleviate current behavior cloning underperformance on more tasks.

[1] https://arxiv.org/abs/2203.03580

---

> ### Author Response · Authors · 2022-10-07
> **Response to the reviewer pzPk [part 1]**
>
> Thank you for your review.
>
> Prompted by TMLR reviews, we added a new section to the revised version of our submission: Section 8 “Limitations and Future work”. We will refer to this section a few times in our response here.
>
> In the following we hope to address all your concerns one by one. If there is anything missing, please let us know.
>
>   > the biggest question in a sense can we even achieve the scale required for effective agents with this approach. While the paper says yes, it doesn't really contribute as much in how except the ability to deal with multi-modal multi-task data under a single framework. However, the data included thousands (if not millions) of episodes on various tasks which is largely only possible with simulations and simulated agents. For the current tasks, the authors likely had access to previously trained checkpoints from existing papers on thee existing benchmarks. However for novel tasks one would require building new simulations as well as figuring out a useful control agent that still gives enough diverse data.
>
> Gato is a data-driven approach, as it is derived from imitation learning. While natural language or image datasets are relatively easy to obtain from the web, a web-scale dataset for control tasks is not currently available. This may seem at first to be problematic, especially when scaling Gato to a higher number of parameters.
>
> That being said, there has already been extensive investigation into this issue. Offline RL aims at leveraging existing control datasets, and its increasing popularity has already resulted in the availability of more diverse and larger datasets. Richer environments and simulations are being built (e.g. Metaverse), and increasing numbers of users already interact with them among thousands of already deployed online games (e.g. there exists a large dataset of Starcraft 2 games). Real-life data has also been already stored for ML research purposes; for example, data for training self-driving cars is acquired from recording human driver data. Finally, while Gato uses data consisting of both observations and corresponding actions, the possibility of using large scale observation-only data to enhance agents has been already studied [Baker et al. 2022]. Thanks to online video sharing and streaming platforms such as Youtube and Twitch, observation-only datasets are not significantly more difficult to collect than natural language datasets, motivating a future research direction to extend Gato to learn from web data.
>
> While the previous paragraph focuses on alleviating drawbacks of data collection from RL agents, it is important to note that this approach presents a different set of tradeoffs compared to scraping web data and can be actually more practical in some situations. Once the simulation is set up and near SOTA agent trained, it can be used to generate massive amounts of high quality data. That is in contrast to the quality of web data which is notorious for its low quality.
>
> In short, we believe that acquiring suitable data is another research question on its own, and this is an active area of research with growing momentum and importance.
>
> We discussed this matter in the new section added to the paper (mentioned in the beginning of our response).
>
>   >  (Robot task it seems had 350k trajectories and still achieved only 50% performance)
>
> We want to stress that the 50% success rate is among the best known for this setting, including specialist agents trained on exactly the same amount of in-domain robotics data using Offline RL (CRR).
>
>   > This is excarberated by the effect that it's unclear whether the "generalist agent" really learned to generalize. Compared to [1] it is a bit surprising that seeing so much data didn't help more in having better representations for finetuning. The evaluated tasks are relatively few and simple and it already had negative transfer on ALE.
>
> We show in Section 5.2 that finetuning of Gato leads to clearly better agents compared to training from scratch for three domains. The improvements are consistent for different numbers of demonstrations used for finetuning. The three domains present different characteristics. Only one domain (ALE Atari) does not benefit when finetuned from Gato.

---

> ### Author Response · Authors · 2022-10-07
> **Response to the reviewer pzPk [part 2]**
>
>
>   > Moreover there wasn't much evaluation on frozen weight zero-shot performance on new tasks as has been demonstrated with various large language models.
>
> Gato is prompted with an expert demonstration, which aids the agent to output actions corresponding to the given task. This is particularly useful since there is otherwise no task identifier available to the agent (that is in contrast to many multi-task RL settings). Gato infers the relevant task from the observations and actions in the prompt.
>
> However, the context length of our agent is limited to 1024 tokens which translates to the agent sometimes attending to only a few environment timesteps in total. This is especially the case for environments with image observations, where depending on the resolution each observation can result in more than one hundred tokens each. Hence for certain environments only a short chunk of a demonstration episode fits in the transformer memory.
>
> Due to this limited prompt context, preliminary experiments with different prompt structures resulted in very similar performance. Similarly, early evaluations of the model using prompt-based in-context learning on new environments did not show a significant performance improvement compared to prompt-less evaluation in the same setting.
>
> Context-length is therefore a current limitation of our architecture, mainly due to the quadratic scaling of self-attention. Many recently proposed architectures enable a longer context at greater efficiency and these innovations could potentially improve our agent performance. We hope to explore these architectures in future work.
>
>   > Could have been useful to analyze how exactly the information from different tasks are being stored in the distributed representations, is it more clustered or more superimposed and whether Mixture-of-Experts style scaling might be even more effective.
>
> We have added a visualization of T-SNE embeddings colorized per task to Section 5.7. As expected, visualizations show that information within a task is clustered together in embedding space, and the language task clusters are closer together.
>
>   > Similarly, would have been interesting to analyze the role of context window for various tasks and whether improving that can alleviate current behavior cloning underperformance on more tasks.
>
> As mentioned above, currently our context length is rather short (please see the aforementioned new section in the paper). Further experimentation with longer contexts is mainly limited by the quadratic scaling of self-attention and our available compute budget. In future work we hope that using more efficient attention architectures can unlock longer context lengths and further improve the performance. Longer memory will also unlock new possibilities such as mentioned by you frozen weight zero-shot prompting.
>
>   > I'm sure there are endless experiments one could do to analyze the nature of representations in the attention layers and various choices one could make for various parameters.
>
> We have shown some attention visualizations in Section 5.6 (and more in Appendix G), and we have also added visualizations of per-task embeddings as mentioned above. Training models to completion can take a significant amount of resources, 256 TPUs for around 5 days, which therefore limits the possible ablations over our hyperparameters that we can present.
>
>   > Unless there are plan on sharing the code base, it would be great to add more concrete details on the finetuning data and settings for each task.
>
> We generated data for the fine-tuning tasks the same way we did for the others (see Section 3.1 for details). Instead of using all the data for a given task, we randomly selected 1000 episodes, then a subset of 100 episodes from the selected episodes, then 10, 5, 3, and finally a single episode. We repeated this procedure 3 times to obtain 3 series of cascading subsets for each task. Each subset is used to conduct one fine-tuning experiment, and each is reported on our plots in Section 5.2 as a separate point.
> We have not altered any of the tasks and used their canonical versions. As 3 out of 4 tasks are open sourced, they do not need further explanation. For the fourth task, DMLab “Order of apples forage simple”, the goal is to collect apples in the right order, green ones first followed by the gold one.
> We extended Appendix E to add all these details.
>
>   > Similarly it would have been great if the paper was more explicit about its limitations even if their hypothesis is "scale is all you need".
>
> We addressed this request by adding the aforementioned new section “Limitations and Future Work”.

---

> ### Author Response · Authors · 2022-10-07
> **Response to the reviewer pzPk [part 3]**
>
>
>   > Appendix C, Table 5: Activation function is supposed to be GeLU?
>
> We used GEGLU activations as stated in the paper. We added a missing reference which caused confusion.
>
>   > Why use AdamW for training but Adam for finetuning? No weight decay?
>
> We found early stopping to be a better regularizer for finetuning in the few-shot setting compared to weight decay. AdamW for pretraining was found to improve results on downstream finetuning.
>
>   > What version of Meta-world? There are quite a few issues in various versions of the benchmark.
>
> It was a version from July 23rd 2021, https://github.com/rlworkgroup/metaworld/commit/a0009ed9a208ff9864a5c1368c04c273bb20dd06. We added this information to the paper.
>
>   > How was the sample weight decided for various datasets?
>
> Initially, we set the domain sampling weight to the square root of the number of tasks in the domain. Then we manually tuned the sampling weights so that no domain was overrepresented.
>
>   > What is the individual dataset sizes in terms of "bytes".
>
> The size in terms of bytes can be estimated from the number of tokens. Each token is 4 bytes and an additional 768 bytes per token for environments with image observations. Using this estimate, one can approximate that the total size of the tokenized control datasets is approximately 150TB.
>
>   > How did you stop training? Were any metrics tracked that can show convergence on any tasks?
>
> We trained for a fixed number of updates (steps), always one million steps. Please see Appendix D for more details.

---

> > ### Comment · Reviewer_pzPk · 2022-10-14
> > **Thank you**
> >
> > Glad you could add more details about the downstream tasks as well as the new section on limitations and future work. I genuinely appreciate the paper's clarity of its point of view!

---

### Review · Reviewer_zvGC · 2022-10-02

**Summary Of Contributions:**

The authors construct a large-scale agent, called Gato, to work as a multi-modal, multi-task, multi-embodiment generalist policy. Specifically, they design a set of tokenization and parameterized embedding functions to unify different input and output forms in varied tasks. After tokenization and embedding, Gato can be trained and sampled from akin to a standard large-scale model. They tested Gato in many scenarios including simulated control tasks, robotics control, and even tasks in the non-policy domain (image captions and chitchat). It verified the hypothesis that training an agent generally capable of a large number of tasks is possible.

**Requested Changes:**

None

**Strengths And Weaknesses:**

Strengths
1. Gato seems to be the agent that can handle most types of tasks in the existing models. The authors validate the agent's skill generalization performance across a sufficient number of scenario types.

2. The description of Gato’s model is well-written, clearly showing its entire construction.

3. The authors provide an exhaustive analysis of the experimental results.

Weaknesses

1. The results of the experiment could be appended with a more refined presentation. There are only detailed results for the specialist Meta-World agent in the appendix, and no detailed results for the generalist agent which could be more conducive to validating the performance of the authors' model and serve as a baseline for later studies.

2. The meaning of Chapter 5.4 is not clear. Compared with Skill Generalization, what conclusion does Skill Mastery, involving the object shapes used for evaluation, want to embody?

3. The description of the DMLAB data in GATO is as follows,

    "We collect data for 255 tasks from the DeepMind Lab, 254 of which are used during training, the left out task was used for out of
    distribution evaluation. Data is collected using an IMPALA agent that has been trained jointly on a set of 18 procedurally generated
    training tasks. Data is collected by executing this agent on each of our 255 tasks, without further training."

    To summarize, there are 18 parent tasks, and 255 tasks derived from these 18 tasks. On 254 of these tasks, data were trained and collected. The remaining one is used for evaluation. The details of these 18 tasks, and the logic of deriving new tasks, are not stated in the paper.

---

> ### Author Response · Authors · 2022-10-07
> **Response to the reviewer zvGC**
>
> Thank you for the encouraging review.
>
> In the following we hope to address all your concerns one by one. If there is anything missing, please let us know.
>
>   > The results of the experiment could be appended with a more refined presentation. There are only detailed results for the specialist Meta-World agent in the appendix, and no detailed results for the generalist agent which could be more conducive to validating the performance of the authors' model and serve as a baseline for later studies.
>
> Initially, we did not report the results, as the final performance of any agent trained via supervised learning depends heavily on the expert performance in the dataset used. As requested, we have added a new section (Appendix L) where we present Gato’s per-domain performance normalized by expert score for future reference.
>
>   > The meaning of Chapter 5.4 is not clear. Compared with Skill Generalization, what conclusion does Skill Mastery, involving the object shapes used for evaluation, want to embody?
>
> The Skill Mastery challenge and the Skill Generalization task are both from the RGB Stacking benchmark and their agent baselines were previously published (Lee, Devin 2021). Skill Mastery is an easier challenge because it evaluates the agent’s ability to grasp objects that were present in the training data (however, the agent still has to generalize to new initial conditions).
>
>   > The description of the DMLAB data in GATO is as follows,
> "We collect data for 255 tasks from the DeepMind Lab, 254 of which are used during training, the left out task was used for out of distribution evaluation. Data is collected using an IMPALA agent that has been trained jointly on a set of 18 procedurally generated training tasks. Data is collected by executing this agent on each of our 255 tasks, without further training."
> To summarize, there are 18 parent tasks, and 255 tasks derived from these 18 tasks. On 254 of these tasks, data were trained and collected. The remaining one is used for evaluation. The details of these 18 tasks, and the logic of deriving new tasks, are not stated in the paper.
>
> The details of these 18 parent tasks and their relation to the rest have been presented at a NeurIPS Workshop in 2021. The link to the presentation will be shared in the camera ready (unanonymised) version.

---

### Decision · Action_Editors · 2022-11-01

**Recommendation:** Accept with minor revision

**Comment:**

All three reviewers were quite positive from the beginning about the quality of this contribution. Reviewers highlighted that the papers convincingly demonstrate a methodology for developing a multi-task/multi-modal agent. The capacities of the agent are evaluated and probed using a set of diverse experiments and analyses. The reviewers also found the paper and descriptions of the model to be clear and well-written. Finally, they mentioned that details were provided that would help other researchers in similar efforts (even if code is not available).

Two of the reviewers made remarks regarding the general methodology developed in the paper and notably that obtaining data at scale for training such an agent was a significant challenge and fundamental limitation of the proposed approach. As a response, the authors added a “Limitations and Future work” Section (8). I found it to be a useful addition that provides food for thought regarding large-scale data acquisition beyond text (e.g. observation + action) for imitation learning and RL and surfaces ideas that others could further develop.

The reviewers each had other minor clarification questions which the authors provided satisfactory replies to. A few led to corrections and clarifications in the paper.

There is one outstanding comment from reviewer zvGC: "To summarize, there are 18 parent tasks, and 255 tasks derived from these 18 tasks. On 254 of these tasks, data were trained and collected. The remaining one is used for evaluation. The details of these 18 tasks, and the logic of deriving new tasks, are not stated in the paper." In their reply, the authors mention that the descriptions of the tasks are currently only available in a non-anonymized format (from a NeurIPS workshop). Since the reviewer could not yet obtain these descriptions, I would ask the authors to provide them as part of their final "camera-ready" version. I recommend acceptance subject to this change.

*Featured Certification.* Two of the three reviewers suggested a "Featured Certification." I do believe that this paper will be of broad interest to our community (even to those that do not work on very large-scale models). It is to my knowledge one of the first to instantiate and show that scaling multi-task and multi-model training can be leveraged to obtain a more general model. It also provides interesting observations regarding forgetting and task interference at scale. Further, and as outlined by reviewers, some of the ingredients it uses for building less narrow models (including tokenization), the diversity of its evaluation, as well as some of the ideas it studies will likely be reused by others. In our private conversation, one of the reviewers mentioned that Gato was "already famous," but I do believe that this paper merits this certification based on its scientific merit alone.

**Audience:**

The paper proposes a new "generalist" agent for solving a variety of tasks. It generalizes ideas of methods recently proposed in large-scale language models. As such, it is of clear interest to the TMLR audience and to the machine-learning research community at large.

**Claims And Evidence:**

Yes, following a round of responses from the authors, the reviewers now are unanimous that the claims in the submission are supported by accurate, convincing, and clear evidence.